# Universal suppression of superfluid weight by disorder independent of quantum geometry and band dispersion

Alexander Lau[ORCID],[1] Sebastiano Peotta[ORCID],[2] Dmitry I. Pikulin[ORCID],[3,4] Enrico Rossi[ORCID],[5] and Timo Hyart[ORCID][1,2]

[1]*International Research Centre MagTop, Institute of Physics,*
*Polish Academy of Sciences, Al. Lotników 32/46, 02-668 Warsaw, Poland*
[2]*Department of Applied Physics, Aalto University, 00076 Aalto, Espoo, Finland*
[3]*Microsoft Quantum, Redmond, Washington 98052, USA*
[4]*Microsoft Quantum, Station Q, Santa Barbara, California 93106-6105, USA*
[5]*Department of Physics, William & Mary, Williamsburg, VA 23187, USA*
(Dated: March 3, 2022)

Motivated by the experimental progress in controlling the properties of the energy bands in superconductors, significant theoretical efforts have been devoted to study the effect of the quantum geometry and the flatness of the dispersion on the superfluid weight. In conventional superconductors, where the energy bands are wide and the Fermi energy is large, the contribution due to the quantum geometry is negligible, but in the opposite limit of flat-band superconductors the superfluid weight originates purely from the quantum geometry of Bloch wave functions. Here, we study how the energy band dispersion and the quantum geometry affect the disorder-induced suppression of the superfluid weight. Surprisingly, we find that the disorder-dependence of the superfluid weight is universal across a variety of models, and independent of the quantum geometry and the flatness of the dispersion. Our results suggest that a flat-band superconductor is as resilient to disorder as a conventional superconductor.

The superfluid weight $D_s$ defines superconductivity, since it captures the ability of a material to sustain a nondissipative current and it becomes nonzero below the critical temperature of the superconductive transition, thereby characterizing the Meissner effect [1–4]. For conventional superconductors originating from the metallic state given by a partially-filled, isolated, and approximately parabolic band, one has the well-known result [1] $D_s = e^2 n/m^*$, where $n$ is the electronic density and $m^*$ the effective mass. Thus, for conventional superconductors the knowledge of the effective mass or, more generally, the band dispersion is sufficient to estimate the superfluid weight.

The observations of superconductivity in twisted bilayer graphene [5–9] and other graphene multilayer systems [10–14] have intensified the theoretical interest in the study of systems with flat energy bands and superconductivity [15–23]. In a flat band the effective mass $m^*$ diverges and one would expect the superfluid weight to vanish. On the contrary, it has been found that, besides the band dispersion, also the quantum geometry of the Bloch wave functions contributes to the superfluid weight [24–33]. In particular, in the case of a well isolated flat band the superfluid weight originates purely from the quantum geometry and can be written as

$$D_s^{\mu\nu} = \frac{8e^2}{\hbar^2}\Delta\sqrt{\nu(1-\nu)}\int\frac{d^dk}{(2\pi)^d}\,g_{\mu\nu}(\mathbf{k}), \qquad (1)$$

where $\Delta$ is the superconducting order parameter, $\nu$ the band filling, $d$ the dimension, and $\mathbf{k}$ the momentum of the electronic state. The quantum metric $g_{\mu\nu}(\mathbf{k})$ is given by the real part of the quantum geometric tensor $\mathcal{B}_{\mu\nu}(\mathbf{k}) = \langle\partial_\mu u_{n\mathbf{k}}|\left(1 - |u_{n\mathbf{k}}\rangle\langle u_{n\mathbf{k}}|\right)|\partial_\nu u_{n\mathbf{k}}\rangle$, where $|u_{n\mathbf{k}}\rangle$ are the Bloch wave functions, $n$ is the band index of the flat band, and $\partial_\mu \equiv \partial_{k_\mu}$. The imaginary part of

$\mathcal{B}_{\mu\nu}(\mathbf{k})$ is proportional to the Berry curvature $B_{\mu\nu}$. Because $\mathcal{B}_{\mu\nu}(\mathbf{k})$ is positive semidefinite, $\mathrm{tr}\, g_{\mu\nu} \geq |B_{12}|$ and $\frac{1}{2}\mathrm{tr}\left[\int d^2k\, g_{\mu\nu}(\mathbf{k})\right] \geq \pi|C|$, where $C = \frac{1}{2\pi}\int d^2k B_{12}(\mathbf{k})$ is the Chern number. This inequality readily translates into a lower bound for the superfluid weight through Eq. (1). Similar lower bounds can be obtained when the bands are characterized by other topological invariants [29].

According to the Anderson theorem, $s$-wave superconductors are robust against perturbations obeying time-reversal symmetry [34]. Therefore, the superconducting ground state can have phase coherence, off-diagonal long-range order, and non-zero superfluid weight even though the underlying single-particle states (and quasiparticle states) are localized due to the disorder [35]. However, by increasing the disorder strength the superconducting order parameter becomes spatially inhomogeneous, its magnitude is suppressed, and finally the system breaks up into superconducting islands separated by regions where the pairing amplitude approximately vanishes [35–37]. As a consequence, the superfluid weight decreases and eventually goes to zero due to quantum phase fluctuations, leading to a superconductor-insulator transition at a critical disorder strength [36–38].

So far these disorder effects have been considered in superconductors where the effect of the quantum geometry is negligible, and the inequalities discussed above between superfluid weight, quantum metric, and Chern number suggest that the disorder effects might be different in flat band systems where the superfluid weight originates from the quantum geometry. In fact, the Chern number is quantized even in the presence of disorder [39], and thus one might expect that the geometric contribution is robust against disorder. On the other hand, the superfluid weight is also affected by the magnitude of the superconducting order parameter, the bandwidth, and

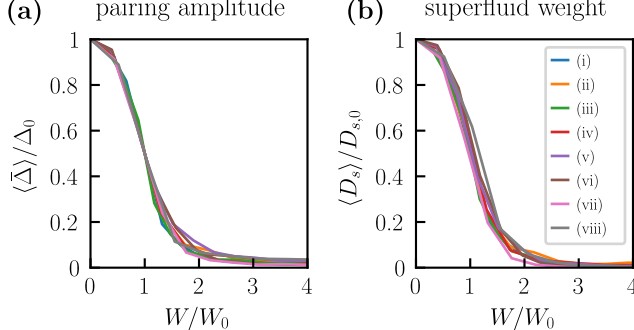

**(a)** pairing amplitude     **(b)** superfluid weight

FIG. 1. Universality of the disorder-induced suppression of the pairing amplitude and the superfluid weight across a variety of lattice models [40]: (i)-(v) topological and trivial extended Kane-Mele models, (vi)-(viii) trivial single-band models (see text and SM [41] for details). The ensemble averages of (a) the spatial average of the pairing amplitude $\bar{\Delta}$ and (b) the superfluid weight $D_s$ are shown as a function of the disorder strength $W/W_0$.

the band gap, which all depend on the disorder strength.

In this Letter, we calculate the disorder-induced suppression of the superfluid weight $D_s$ for a generalization of the Kane-Mele model [42], for which the low energy bands' topology and flatness can be easily tuned by varying the values of the model's parameters, and for a simple single band model. Our main results are shown in Fig. 1, where the ensemble averages $\langle\bar{\Delta}\rangle/\Delta_0$, $\langle D_{\rm s}\rangle/D_{\rm s,0}$ of the spatially averaged pairing potential $\bar{\Delta}$ and the superfluid weight $D_{\rm s}$ are shown as a function of the disorder strength $W/W_0$. $\Delta_0$, $D_{\rm s,0}$ are the pairing potential and the superfluid weight, respectively, in the clean limit. $W_0$ is defined as the value of $W$ for which $\langle\bar{\Delta}\rangle/\Delta_0 = 1/2$. For $W \approx W_0$ the superconductor breaks up into superconducting islands. In all models the disorder dependence of $\langle\bar{\Delta}\rangle$ and $\langle D_{\rm s}\rangle$ is the same after rescaling, pointing to an unexpected universal behavior.

We consider a variety of tight-binding Hamiltonians $H_0$ with disorder potentials $V_d$ supplemented by the pairing interaction $H_{\rm int}$, so that the full Hamiltonian is $H = H_0 + H_{\rm int} + V_d$. We assume that $H_0$ obeys $U(1)$ spin-rotation symmetry, $V_d$ is represented by uncorrelated on-site energies uniformly distributed in the interval $[-W, W]$, and $H_{\rm int}$ describes a local attraction of strength $U$ between the electrons that leads to a time-reversal invariant singlet superconducting state described by a real-valued pairing potential $\Delta(\mathbf{r})$. We neglect the frequency dependence of $U$ and the renormalization of $U$ due to the localization, because we are interested in the comparison of the models instead of seeking for a quantitative description of a particular system.

To model the disorder potential, we consider a large cluster of $N$ sites repeated in space $\mathcal{N}$ times with periodic boundary conditions. The full set of superconducting

mean-field equations for such a system is given by

$$\Delta_\alpha = \frac{1}{\mathcal{N}} \sum_i U \langle c_{i\alpha\uparrow} c_{i\alpha,\downarrow}\rangle, \ \nu = \frac{1}{N\mathcal{N}} \sum_{i,\alpha,\sigma} \langle c_{i\alpha\sigma}^\dagger c_{i\alpha\sigma}\rangle \ (2)$$

with $\alpha = 1, \dots N$, $U > 0$, and the filling per lattice site $\nu \in [0,2]$ associated with both spin channels $\sigma = \uparrow, \downarrow$ [see Supplemental Material (SM) [41]]. The operators $c_{i\alpha\sigma}^\dagger$ ($c_{i\alpha\sigma}$) create (annihilate) an electron with spin $\sigma$ at site $\mathbf{r}_\alpha$ in the $i$-th cluster. This is a large set of $N+1$ equations, which we have to solve self-consistently for the chemical potential $\mu$ and for the spatial profile of the superconducting order parameter $\Delta_\alpha$ at a given temperature $T$ and interaction strength $U$. Therefore, to reduce the computational cost of the calculation of $\Delta(T, \mathbf{r}_\alpha)$, we assume that the spatial profile is approximately independent of temperature. With this assumption, we obtain the (normalized) spatial profile $\hat{\Delta}(\mathbf{r}_\alpha)$ from the linearized self-consistency equations, which are valid close to the critical temperature, and the overall amplitude $\|\Delta(T)\| \equiv [\sum_\alpha |\Delta(T, \mathbf{r}_\alpha)|^2]^{1/2}$ and $\nu$ from the nonlinear self-consistency equations (see SM [41]) to obtain $\Delta(T, \mathbf{r}_\alpha) = \|\Delta(T)\|\hat{\Delta}(\mathbf{r}_\alpha)$. We find that this approximation leads to an underestimation of $\langle\bar{\Delta}\rangle$ that, being very similar for all the models (see SM [41]), does not affect the relative comparison of the models.

Given a specific disorder realization, we compute the corresponding superconducting order parameter $\Delta_\alpha$ and the chemical potential self-consistently employing the reduced mean-field equations, and diagonalize the associated Bogoliubov-de Gennes Hamiltonian $H_{\rm BdG}$ to determine its excitation energies $E_i(\mathbf{k})$ and eigenstates $\psi_i(\mathbf{k})$, where $\mathbf{k}$ is the superlattice momentum arising due to the cluster periodicity and $i$ is the band index. The full superfluid weight $D_s$ of the superconductor is given by

$$D_s^{\mu\nu} = \frac{e^2}{\hbar^2} \sum_{\mathbf{k},ij} \frac{n(E_j) - n(E_i)}{E_i - E_j} \left( \langle\partial_\mu H_{\rm BdG}\rangle_{ij} \langle\partial_\nu H_{\rm BdG}\rangle_{ji} \right.$$
$$\left. - \langle\partial_\mu H_{\rm BdG}\gamma^z\rangle_{ij} \langle\gamma^z \partial_\nu H_{\rm BdG}\rangle_{ji} \right), \quad (3)$$

where $\langle\cdot\rangle_{ij} \equiv \langle\psi_i| \cdot |\psi_j\rangle$, $n(E_i)$ is the Fermi function, and $\gamma^z = \sigma_z \otimes \mathbb{1}_{N\times N}$ with $\sigma_z$ being a Pauli matrix in particle-hole space (see SM [41]). We further decompose the full superfluid weight into a conventional contribution $D_{s,\rm conv}$ and a geometric contribution $D_{s,\rm geom}$. The conventional contribution involves only intraband matrix elements containing derivatives of the normal-state Hamiltonian's energies $\epsilon_{\mathbf{k}m\sigma}$,

$$D_{s,\rm conv}^{\mu\nu} = \sum_{\mathbf{k},mp} C_{pp}^{mm} \left[\partial_\mu \epsilon_{\mathbf{k}m\uparrow} \partial_\nu \epsilon_{-\mathbf{k},p,\downarrow} + \mu \leftrightarrow \nu\right], \ (4)$$

with coefficients $C_{pp}^{mm}$ given in the SM [41]. The geometric contribution, $D_{s,\rm geom}$, comprises interband matrix elements with derivatives of the normal-state Hamiltonian's Bloch states (see SM [41]) and can be obtained as the difference between $D_s$ and $D_{s,\rm conv}$. In the limits

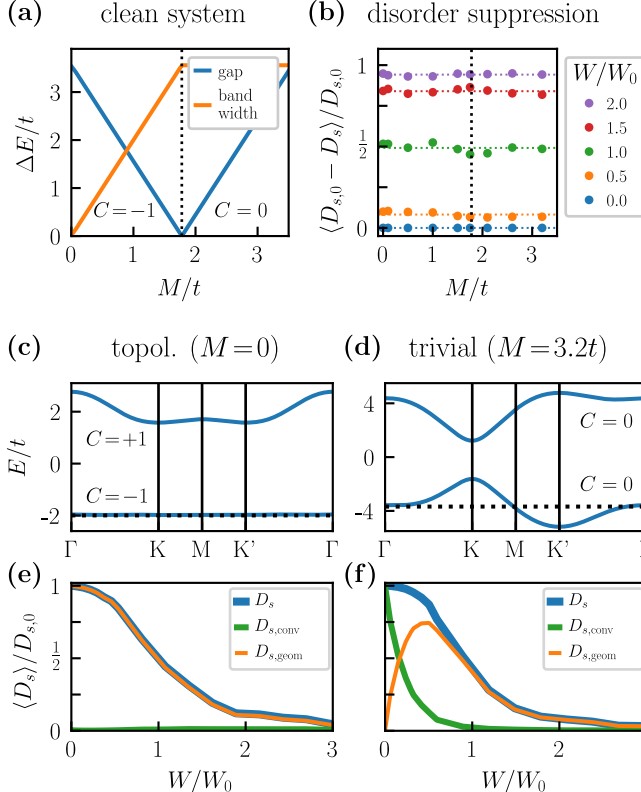

FIG. 2. Disorder-induced suppression of the superfluid weight in the extended Kane-Mele model [40]. (a) Evolution of the energy gap and the bandwidth of the lower band as a function of $M$. $C$ is the Chern number of the lower spin-up band. (b) $D_s$ as a function of $M$ for different values of $W/W_0$ and $\nu = 1/2$. The vertical dotted black line indicates the topological transition in the clean system. (c), (d) Energy bands of the clean systems along high-symmetry lines of the Brillouin zone for $M$ values corresponding to topologically distinct cases. The dotted black lines indicate the Fermi level corresponding to $\nu = 1/2$. (e), (f) $\langle D_s \rangle$ as a function of $W/W_0$ for $\nu = 1/2$.

of a trivial parabolic band and an ideal flat band without disorder, this decomposition reproduces the conventional result $D_s = e^2 n/m^*$, and Eq. (1), respectively. The considered disorder preserves the symmetries of the respective clean systems on average. In particular, on average it preserves the $C_3$ symmetry of the Kane-Mele models and the $C_4$ symmetry of the single-band models (see SM [41]). Consequently, the disorder-averaged superfluid weight tensors of our models are proportional to the identity matrix and we have $D_s^{xx} = D_s^{yy} \equiv D_s$.

We first consider an extended Kane-Mele model on a honeycomb lattice given by a Haldane model [43] for each spin channel and additional hoppings between 3rd- ($t_3$)

and 4th-nearest neighbors ($t_4$):

$$
\begin{aligned}
H = {} & t \sum_{\sigma,\langle i,j \rangle_1} c_{j\sigma}^\dagger c_{i\sigma} + t_2 \sum_{\sigma,\langle i,j \rangle_2} e^{i\sigma\varphi_{ij}} c_{j\sigma}^\dagger c_{i\sigma} + t_3 \sum_{\sigma,\langle i,j \rangle_3} c_{j\sigma}^\dagger c_{i\sigma} \\
& + t_4 \sum_{\sigma,\langle i,j \rangle_4} c_{j\sigma}^\dagger c_{i\sigma} + \sum_{\sigma,i} \left[ (-1)^i M - \mu \right] c_{i\sigma}^\dagger c_{i\sigma},
\end{aligned}
\tag{5}
$$

Here, $\langle i,j \rangle_n$ denotes pairs of $n$-th neighbors, $\sigma = \pm 1 \equiv \uparrow, \downarrow$ is the spin index of the particles, $M$ is a staggered on-site potential, $\mu$ is the chemical potential, and $\varphi_{ij} = \pm\varphi$ is a next-nearest-neighbor (NNN) hopping phase whose sign depends on the hopping direction and on the spin (see SM [41]). The spin-dependence of the NNN hopping phase is chosen in such a way that the full non-interacting Hamiltonian is time-reversal symmetric. We call the model in Eq. (5) the extended Kane-Mele model because in the limit $t_3 = t_4 = 0$ and $\varphi = \pi/2$ it reduces to the model introduced by Kane and Mele in Ref. 42.

Importantly, our model is well-suited for the study of topological flat bands: By taking $t_2 = 0.349t$, $t_3 = -0.264t$, $t_4 = 0.026t$, $\varphi = 1.377$, and $M = 0$ [model (i) in Fig. 1], the lowest spin-degenerate bands are almost flat and have Chern numbers $C = \pm 1$ [see Fig. 2(a), (c)]. Therefore the superfluid weight is almost entirely geometric in the clean limit, i.e., $D_s \simeq D_{s,\mathrm{geom}}$ satisfying Eq. (1) with $\Delta \approx U\sqrt{\nu(1-\nu)}/2$. Fig. 2(e) shows the disorder-averaged superfluid weight $\langle D_s \rangle$ for $\nu = 1/2$, $U = 3t$ and $T = 0$ [44] displaying the behavior already presented in Fig. 1(b), but here we have decomposed it into geometric and conventional contributions [45]. The superfluid weight associated with a flat band is almost entirely geometric for all values of the disorder strength.

By increasing $M$ the previously flat band becomes more dispersive and the bulk energy gap closes around $M = 1.75t$, so that after the reopening of the bulk gap both energy bands are trivial ($C = 0$) [Fig. 2(a)]. Thus, as we increase the parameter $M$, the superfluid weight acquires a finite conventional contribution due to the growing dispersion of the lower band. The fraction of the geometric contribution decreases, so that deep inside the trivial phase the geometric contribution practically vanishes and the superfluid weight becomes almost entirely conventional in the absence of disorder. This picture changes with increasing disorder, as we show in Fig. 2(f). First, we observe that the conventional contribution is linearly suppressed in the low-disorder regime, whereas the suppression is quadratic for the full superfluid weight. In contrast, the geometric contribution is enhanced for small disorder until it reaches a turning point. At this point the conventional contribution is nearly zero and the superfluid weight becomes entirely geometric, even though the underlying bands are topologically trivial.

Importantly, although the geometric and conventional contributions are remarkably different depending on $M$, surprisingly the disorder induced suppression of the scaled superfluid weight $\langle D_s \rangle/D_0$ as a function of the scaled disorder strength $W/W_0$ is completely independent of the value of $M$ [see Fig. 2(b)]. We find essentially

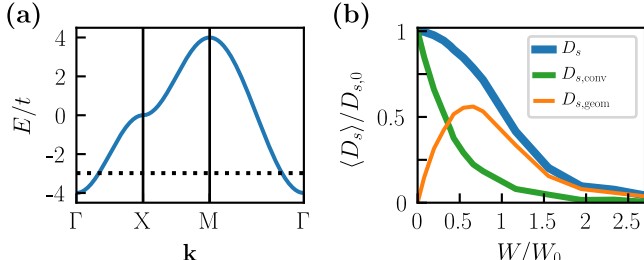

FIG. 3. Disorder-induced suppression of $\langle D_s \rangle$ in a single-band model [40]. (a) Energy band dispersion of the clean system along high-symmetry lines of the Brillouin zone. The dotted line indicates the Fermi level at $\nu = 1/5$. (b) $\langle D_s \rangle$, $\langle D_{s,\text{conv}} \rangle$ and $\langle D_{s,\text{geom}} \rangle$ as a function of $W$ at $\nu = 1/5$.

the same results also when other parameters are varied, such as the NNN hopping phase $\varphi$ [see SM [41] and model (iv) in Fig. 1].

It is instructive to compare the extended Kane-Mele model with a nearest-neighbor tight-binding model on the square lattice given by the Hamiltonian

$$H = -t \sum_{\sigma, \langle i,j \rangle} c_{j\sigma}^\dagger c_{i\sigma} - \mu \sum_{\sigma, i} c_{i\sigma}^\dagger c_{i\sigma}. \qquad (6)$$

The energy spectrum consists of only one band with the dispersion relation $E(k_k, k_y) = -2t(\cos k_x + \cos k_y)$ visualized in Fig. 3(a). The filling is set at $\nu = 1/5$. In the clean limit, the Berry curvature, the Chern number, and the geometric contribution are identically zero, and $D_s = D_{s,\text{conv}} \approx e^2 n/m^* = 2e^2 \nu t/\hbar^2$. However, with the onset of disorder the geometric contribution is again enhanced while the conventional contribution is linearly suppressed [Fig. 3(b)]. In particular, $D_s$ is entirely geometric in the strong disorder regime, even though the considered model in the clean limit has no geometric structure.

A possible explanation for this counter-intuitive behavior lies in the nature of the decomposition of the superfluid weight. By definition, the geometric part contains only interband matrix elements (see SM [41]). Hence, it is identically zero for a single-band model in the clean limit. By adding disorder to the system, this band fans out into several subbands in superlattice mini Brillouin zone arising from the cluster periodicity. As a consequence, states previously separated in momentum space may now couple giving rise to nonzero interband matrix elements. Moreover, avoided crossings in the superlattice Brillouin zone can act as hot-spots of quantum metric. In the thermodynamic limit $(N \to \infty)$, the superlattice Brillouin zone collapses to a single point allowing all states of the single band to couple. Consequently, the superfluid weight now originates entirely from interband terms so that only the geometric contribution of the superfluid weight remains non-zero.

Evidently, the meaning of the geometric and conventional contributions is obscured in the case of dirty su-

perconductors. To understand this better it is important to compare the disorder-induced suppression of the total superfluid weight in the cases of the extended Kane-Mele models (Fig. 2) and the single-band model (Fig. 3). As shown in Fig. 1, the scaled superfluid weight $\langle D_s \rangle / D_0$ as a function of the scaled disorder strength $W/W_0$ behaves the same way in all models. This finding is independent of the concrete decomposition of the superfluid weight into conventional and geometric contributions. In particular, it would still hold even if there existed a different decomposition for disordered systems. Thus, our results imply that the microscopic mechanism underlying the superfluid weight becomes unimportant in dirty superconductors.

To summarize, we have demonstrated that the disorder-induced suppression of the superfluid weight is universal across a variety of theoretical models independently of the quantum geometry and the flatness of the dispersion. Thus, flat-band superconductors are as resilient to disorder as conventional superconductors. We have mainly concentrated on the disorder-induced suppression of the ensemble averages of the pairing potential and the superfluid weight. However, the universality across the models remains true also for the statistical fluctuations. Namely, we find that apart from the transition regime $W \approx W_0$, also the standard deviations $\sigma(\bar{\Delta})/\Delta_0$ and $\sigma(D_s)/D_{s,0}$ as a function of $W/W_0$ behave the same way in all models (see SM [41]). In our calculation there is no critical value of $W$ above which $D_s$ vanishes. This is due to the fact that in our approach relative phase fluctuations of $\Delta$ between different superconducting regions (islands) of the inhomogenous landscape induced by the disorder are not taken into account [36–38]. The interplay of such fluctuations and the quantum metric is an interesting direction for future research.

Graphene-based heterostructures are an ideal platform to experimentally study the universality of the disorder-induced suppression of the superfluid weight. These systems are intrinsically very clean, disorder can be introduced in a controlled way, and it is possible to tune the dispersion and the different contributions of the superfluid weight through the twist angle, pressure, and electric field [5, 6, 10, 11, 28, 30]. Recent experiments indicate that in some graphene-based systems it might be possible to realize unconventional superconducting order parameters [12–14]. Therefore, it is an interesting direction for future research to find out if the disorder-induced suppression of superfluid weight remains independent of the quantum geometry beyond the time-reversal invariant s-wave superconductors considered in this work.

*Acknowledgments:* The research was partially supported by the Foundation for Polish Science through the IRA Programme co-financed by EU within SG OP. A. L. acknowledges support from a Marie Skłodowska-Curie Individual Fellowship under grant MagTopCSL (ID 101029345). E.R. acknowledges support from DOE, Grant No. DE-SC0022245. S. P. acknowledges support from the Academy of Finland under Grants No. 330384

and No. 336369. We acknowledge the computational resources provided by the Aalto Science-IT project and the access to the computing facilities of the Interdisciplinary Center of Modeling at the University of Warsaw, Grant No. G84-4, G87-1135, and G88-1203.

*Data availability:* The data shown in the figures and the code generating all of the data is available at Ref. 46.

[1] D. J. Scalapino, S. R. White, and S. C. Zhang, Superfluid density and the Drude weight of the Hubbard model, Physical Review Letters **68**, 2830 (1992).

[2] D. J. Scalapino, S. R. White, and S. Zhang, Insulator, metal, or superconductor: The criteria, Physical Review B **47**, 7995 (1993).

[3] V. L. Berezinskiĭ, Destruction of Long-range Order in One-dimensional and Two-dimensional Systems having a Continuous Symmetry Group I. Classical Systems, Soviet Journal of Experimental and Theoretical Physics **32**, 493 (1971).

[4] J. M. Kosterlitz and D. J. Thouless, Ordering, metastability and phase transitions in two-dimensional systems, Journal of Physics C: Solid State Physics **6**, 1181 (1973).

[5] Y. Cao, V. Fatemi, S. Fang, K. Watanabe, T. Taniguchi, E. Kaxiras, and P. Jarillo-Herrero, Unconventional superconductivity in magic-angle graphene superlattices, Nature **556**, 43 (2018).

[6] M. Yankowitz, S. Chen, H. Polshyn, Y. Zhang, K. Watanabe, T. Taniguchi, D. Graf, A. F. Young, and C. R. Dean, Tuning superconductivity in twisted bilayer graphene, Science **363**, 1059 (2019).

[7] X. Lu, P. Stepanov, W. Yang, M. Xie, M. A. Aamir, I. Das, C. Urgell, K. Watanabe, T. Taniguchi, G. Zhang, A. Bachtold, A. H. MacDonald, and D. K. Efetov, Superconductors, orbital magnets and correlated states in magic-angle bilayer graphene, Nature **574**, 653 (2019).

[8] P. Stepanov, I. Das, X. Lu, A. Fahimniya, K. Watanabe, T. Taniguchi, F. H. L. Koppens, J. Lischner, L. Levitov, and D. K. Efetov, Untying the insulating and superconducting orders in magic-angle graphene, Nature **583**, 375 (2020).

[9] Y. Saito, J. Ge, K. Watanabe, T. Taniguchi, and A. F. Young, Independent superconductors and correlated insulators in twisted bilayer graphene, Nature Physics **16**, 926 (2020).

[10] J. M. Park, Y. Cao, K. Watanabe, T. Taniguchi, and P. Jarillo-Herrero, Tunable strongly coupled superconductivity in magic-angle twisted trilayer graphene, Nature **590**, 249 (2021).

[11] Z. Hao, A. M. Zimmerman, P. Ledwith, E. Khalaf, D. H. Najafabadi, K. Watanabe, T. Taniguchi, A. Vishwanath, and P. Kim, Electric field–tunable superconductivity in alternating-twist magic-angle trilayer graphene, Science **371**, 1133 (2021).

[12] H. Zhou, T. Xie, T. Taniguchi, K. Watanabe, and A. F. Young, Superconductivity in rhombohedral trilayer graphene, Nature **598**, 434 (2021).

[13] Y. Cao, J. M. Park, K. Watanabe, T. Taniguchi, and P. Jarillo-Herrero, Pauli-limit violation and re-entrant superconductivity in moiré graphene, Nature **595**, 526 (2021).

[14] H. Zhou, L. Holleis, Y. Saito, L. Cohen, W. Huynh, C. L. Patterson, F. Yang, T. Taniguchi, K. Watanabe, and A. F. Young, Isospin magnetism and spin-polarized superconductivity in Bernal bilayer graphene, Science **0**, eabm8386 (2022).

[15] E. Suárez Morell, J. D. Correa, P. Vargas, M. Pacheco, and Z. Barticevic, Flat bands in slightly twisted bilayer graphene: Tight-binding calculations, Physical Review B **82**, 121407 (2010).

[16] R. Bistritzer and A. H. MacDonald, Moire bands in twisted double-layer graphene, Proceedings of the National Academy of Sciences **108**, 12233 (2011).

[17] N. B. Kopnin, T. T. Heikkilä, and G. E. Volovik, High-temperature surface superconductivity in topological flat-band systems, Phys. Rev. B **83**, 220503 (2011).

[18] R. Ojajärvi, T. Hyart, M. A. Silaev, and T. T. Heikkilä, Competition of electron-phonon mediated superconductivity and Stoner magnetism on a flat band, Phys. Rev. B **98**, 054515 (2018).

[19] T. J. Peltonen, R. Ojajärvi, and T. T. Heikkilä, Mean-field theory for superconductivity in twisted bilayer graphene, Phys. Rev. B **98**, 220504 (2018).

[20] F. Wu, A. H. MacDonald, and I. Martin, Theory of Phonon-Mediated Superconductivity in Twisted Bilayer Graphene, Physical Review Letters **121**, 257001 (2018).

[21] A. Lau, T. Hyart, C. Autieri, A. Chen, and D. I. Pikulin, Designing Three-Dimensional Flat Bands in Nodal-Line Semimetals, Phys. Rev. X **11**, 031017 (2021).

[22] Y.-Z. Chou, F. Wu, J. D. Sau, and S. Das Sarma, Acoustic-Phonon-Mediated Superconductivity in Rhombohedral Trilayer Graphene, Phys. Rev. Lett. **127**, 187001 (2021).

[23] Y.-Z. Chou, F. Wu, J. D. Sau, and S. D. Sarma, Acoustic-phonon-mediated superconductivity in Bernal bilayer graphene (2021), arXiv:2110.12303 [cond-mat.supr-con].

[24] K. Moon, H. Mori, K. Yang, S. M. Girvin, A. H. MacDonald, L. Zheng, D. Yoshioka, and S.-C. Zhang, Spontaneous interlayer coherence in double-layer quantum Hall systems: Charged vortices and Kosterlitz-Thouless phase transitions, Phys. Rev. B **51**, 5138 (1995).

[25] S. Peotta and P. Törmä, Superfluidity in topologically nontrivial flat bands, Nature Communications **6**, 8944 (2015).

[26] L. Liang, T. I. Vanhala, S. Peotta, T. Siro, A. Harju, and P. Törmä, Band geometry, Berry curvature, and superfluid weight, Phys. Rev. B **95**, 024515 (2017).

[27] T. Hazra, N. Verma, and M. Randeria, Bounds on the Superconducting Transition Temperature : Applications to Twisted Bilayer Graphene and Cold Atoms, Physical Review X **9**, 31049 (2019).

[28] X. Hu, T. Hyart, D. I. Pikulin, and E. Rossi, Geometric and conventional contribution to the superfluid weight in twisted bilayer graphene, Phys. Rev. Lett. **123**, 237002 (2019).

[29] F. Xie, Z. Song, B. Lian, and B. A. Bernevig, Topology-Bounded Superfluid Weight in Twisted Bilayer Graphene, Phys. Rev. Lett. **124**, 167002 (2020).

[30] A. Julku, T. J. Peltonen, L. Liang, T. T. Heikkilä,

and P. Törmä, Superfluid weight and Berezinskii-Kosterlitz-Thouless transition temperature of twisted bilayer graphene, Phys. Rev. B **101**, 060505 (2020).

[31] X. Hu, T. Hyart, D. I. Pikulin, and E. Rossi, Quantum-metric-enabled exciton condensate in double twisted bilayer graphene (2020), arXiv:2008.03241 [cond-mat.mes-hall].

[32] E. Rossi, Quantum metric and correlated states in two-dimensional systems, Current Opinion in Solid State and Materials Science **25**, 100952 (2021).

[33] P. Törmä, S. Peotta, and B. A. Bernevig, Superfluidity and Quantum Geometry in Twisted Multilayer Systems (2021), arXiv:2111.00807 [cond-mat.supr-con].

[34] P. Anderson, Theory of dirty superconductors, Journal of Physics and Chemistry of Solids **11**, 26 (1959).

[35] M. Ma and P. A. Lee, Localized superconductors, Phys. Rev. B **32**, 5658 (1985).

[36] A. Ghosal, M. Randeria, and N. Trivedi, Role of spatial amplitude fluctuations in highly disordered $s$-wave superconductors, Phys. Rev. Lett. **81**, 3940 (1998).

[37] A. Ghosal, M. Randeria, and N. Trivedi, Inhomogeneous pairing in highly disordered s-wave superconductors, Phys. Rev. B **65**, 014501 (2001).

[38] M. Ma, B. I. Halperin, and P. A. Lee, Strongly disordered superfluids: Quantum fluctuations and critical behavior, Phys. Rev. B **34**, 3136 (1986).

[39] J. Bellissard, A. van Elst, and H. Schulz- Baldes, The noncommutative geometry of the quantum Hall effect, Journal of Mathematical Physics **35**, 5373 (1994).

[40] For the results shown in Figs. 1 and 2(b), we used cluster sizes $N = 128$ for the Kane-Mele models and $N = 121$ for the single-band models. We calculated the ensemble averages over 30 disorder realizations. For the decomposition of the superfluid weight into conventional and geometric part, shown in Figs. 2(e,f) and 3(b), we used $N = 50$ for the Kane-Mele models and $N = 49$ for the single-band models. We calculated the ensemble averages

over 40 disorder realizations.

[41] See Supplemental Material at [. . . ], which includes Refs. [47, 48], for the derivation of the self-consistent mean-field equations, for additional details of the superfluid weight and its decomposition, for more details on the extended Kane-Mele models, for a discussion of standard deviations, for an analysis of the size scaling of the disorder-induced suppression, for a comparison of solutions from the full and from the reduced mean-field equations, for more details on the single-band models, and for additional details on the superfluid weight of the clean systems.

[42] C. L. Kane and E. J. Mele, Quantum Spin Hall Effect in Graphene, Phys. Rev. Lett. **95**, 226801 (2005).

[43] F. D. M. Haldane, Model for a Quantum Hall Effect without Landau Levels: Condensed-Matter Realization of the "Parity Anomaly", Phys. Rev. Lett. **61**, 2015 (1988).

[44] In the numerics, the $T = 0$ limit is approximated by $T \leq T_c/100$.

[45] Since we use a smaller cluster size to obtain the data shown in Fig. 2(e) and (f), the superfluid weight for strong disorder ($W/W_0 > 2$) is on average larger compared to Fig. 1(b). Finite-size effects are discussed in the SM [41].

[46] A. Lau, S. Peotta, D. I. Pikulin, E. Rossi, and T. Hyart, Universal suppression of superfluid weight by disorder independent of quantum geometry and band dispersion, zenodo.6012755 10.5281/zenodo.6012755 (2022).

[47] C. W. Groth, M. Wimmer, A. R. Akhmerov, and X. Waintal, Kwant: a software package for quantum transport, New J. Phys. **16**, 063065 (2014).

[48] T. Fukui, Y. Hatsugai, and H. Suzuki, Chern numbers in discretized Brillouin zone: Efficient method of computing (spin) Hall conductances, Journal of the Physical Society of Japan **74**, 1674 (2005).

# SUPPLEMENTAL MATERIAL

## CONTENTS

## I. SELF-CONSISTENT MEAN-FIELD EQUATIONS

We start from the generic form of a Hamiltonian for a singlet s-wave superconductor with time-reversal symmetry,

$$H_{BdG} = \sum_{ij} H_{ij} \, c_{i\uparrow}^{\dagger} c_{j\uparrow} - \sum_{ij} H_{ij} \, c_{i\downarrow} c_{j\downarrow}^{\dagger} + \sum_i [\Delta_i \, c_{i\uparrow}^{\dagger} c_{i\downarrow}^{\dagger} + h.c.], \tag{S1}$$

where we have used that $H_{\uparrow} = H_{\downarrow}^T \equiv H$ because of time-reversal symmetry. The doubling of degrees of freedom in the BdG formalism has been removed by making use of the relation between spin-up electrons and spin-down holes due to time-reversal symmetry and $U(1)$ spin rotation symmetry. In general, the indices $i, j$ may include all degrees of freedom except the spin. Here, we focus on models with only a site degree of freedom, i.e., one orbital per site. Hence, the operator $c_{i\sigma}^{\dagger}$ creates a particle with spin $\sigma$ at site $r_i$.

Let us assume the system is periodic in space with $N$ sites per unit cell. For a lattice site at position $r$, we then write $r = R_i + r_\alpha$, where $R_i$ points to the origin of the $i$-th unit cell and $r_\alpha$ is the position of the lattice site inside that unit cell. The index $\alpha$ now enumerates the lattice sites within a unit cell. With this, we rewrite the BdG Hamiltonian as follows

$$H_{BdG} = \sum_{ij} \sum_{\alpha,\beta} H_{i\alpha,j\beta} \, c_{i\alpha\uparrow}^{\dagger} c_{j\beta\uparrow} - \sum_{ij} \sum_{\alpha\beta} H_{i\alpha,j\beta} \, c_{i\alpha\downarrow} c_{j\beta\downarrow}^{\dagger} + \sum_{i,\alpha} [\Delta_\alpha \, c_{i\alpha\uparrow}^{\dagger} c_{i\alpha\downarrow}^{\dagger} + h.c.], \tag{S2}$$

where we have assumed that the superconducting pairing amplitude has the same translational symmetry as the normal-state Hamiltonian, i.e., $\Delta_{i\alpha} \equiv \Delta_\alpha \, \forall i$. Making use of the periodicity, we apply a Fourier transformation of the following form

$$c_{i\alpha\sigma}^{\dagger} = \frac{1}{\sqrt{\mathcal{N}}} \sum_k e^{ik(R_i + r_\alpha)} c_{k\alpha\sigma}^{\dagger}, \tag{S3}$$

where $\mathcal{N}$ is the total number of unit cells in the system, and obtain

$$H_{BdG} = \sum_k \sum_{\alpha\beta} H_{\alpha\beta}(k)\, c_{k\alpha\uparrow}^\dagger c_{k\beta\uparrow} - \sum_k \sum_{\alpha\beta} H_{\alpha\beta}(k)\, c_{-k,\alpha,\downarrow} c_{-k,\beta,\downarrow}^\dagger + \sum_{k,\alpha}[\Delta_\alpha\, c_{k\alpha\uparrow}^\dagger c_{-k,\alpha\downarrow}^\dagger + h.c.], \tag{S4}$$

where the components of the normal-state Bloch Hamiltonian are defined as

$$H_{\alpha\beta}(k) = \sum_\delta t_{\alpha\beta}(\delta)e^{ik(R_\delta + r_\alpha - r_\beta)}, \ \text{with } t_{\alpha\beta}(\delta) \equiv H_{i+\delta,\alpha;i\beta}\,\forall i. \tag{S5}$$

We assume that superconductivity originates from an attractive on-site interaction, such that the mean-field equations can be written as

$$\Delta_\alpha = \frac{1}{\mathcal{N}}\sum_i \Delta_{i\alpha} = \frac{1}{\mathcal{N}}\sum_i U\langle c_{i\alpha\uparrow} c_{i\alpha,\downarrow}\rangle = \frac{U}{\mathcal{N}}\sum_k \langle c_{k\alpha\uparrow} c_{-k,\alpha,\downarrow}\rangle, \tag{S6}$$

with the paring interaction $U > 0$.

## A. Full set of mean-field equations

We introduce the Nambu-space vector $C_k^\dagger = [c_{k,1,\uparrow}^\dagger, \ldots, c_{k,N,\uparrow}^\dagger, c_{-k,1,\downarrow}, \ldots, c_{-k,N,\downarrow}]$ and write the BdG Hamiltonian as follows

$$H_{BdG} = \sum_k \left[ C_k^\dagger H^0(k)\, C_k + C_k^\dagger H^1(k)\, C_k \right], \tag{S7}$$

with

$$H^0(k) = \begin{pmatrix} h_0(k) & 0 \\ 0 & -h_0(k) \end{pmatrix} \tag{S8}$$

and

$$H^1(k) = H^1 = \begin{pmatrix} 0 & \mathrm{diag}\,(\Delta_1, \ldots, \Delta_N) \\ \mathrm{diag}\,(\Delta_1^*, \ldots, \Delta_N^*) & 0 \end{pmatrix}, \tag{S9}$$

where $h_0(k)$ is the normal-state Bloch Hamiltonian with components given by $H_{\alpha\beta}(k)$ from Eq. (S5). We introduce new creation operators defined by

$$d_{ak}^\dagger = \sum_\alpha \left[ \Psi_{a\alpha,+}(k)\, c_{k\alpha\uparrow}^\dagger + \Psi_{a\alpha,-}(k)\, c_{-k,\alpha,\downarrow} \right], \tag{S10}$$

or, equivalently,

$$c_{k\alpha\uparrow}^\dagger = \sum_a \Psi_{a\alpha,+}^*(k)\, d_{ak}^\dagger, \tag{S11}$$

$$c_{-k,\alpha,\downarrow} = \sum_a \Psi_{a\alpha,-}^*(k)\, d_{ak}^\dagger. \tag{S12}$$

Here, we have $a = 1, \ldots, 2N$ and $[\Psi_{a,+}(k), \Psi_{a,-}(k)]^T$ are eigenvectors of $H^0(k) + H^1(k)$ with energy $E_a(k)$. We have introduced subscripts $\pm$ to distinguish matrix elements associated with operators $c_{k\alpha\uparrow}^\dagger$ (+) and $c_{-k\alpha\downarrow}$ (−) of the Nambu-space vector. With this, the BdG Hamiltonian becomes

$$H_{BdG} = \sum_{a,k} E_a(k)\, d_{ak}^\dagger d_{ak}. \tag{S13}$$

Similarly, we get

$$\langle c_{k\alpha\uparrow} c_{-k,\alpha,\downarrow}\rangle = \sum_a \Psi_{a\alpha,+}(k)\Psi_{a\alpha,-}^*(k)\langle d_{ak} d_{ak}^\dagger\rangle = \sum_a \Psi_{a\alpha,+}(k)\Psi_{a\alpha,-}^*(k)\{1 - n_F[E_a(k)]\}. \tag{S14}$$

Hence, the self-consistency equation from Eq. (S6) becomes,

$$\Delta_\alpha = \frac{U}{\mathcal{N}} \sum_{k,a} \Psi_{a\alpha,+}(k)\Psi^*_{a\alpha,-}(k)\{1 - n_F[E_a(k)]\}, \tag{S15}$$

with the Fermi function $n_F(E) = (e^{\beta E} + 1)^{-1}$. This is a set of $N$ equations for $(\Delta_1, \ldots, \Delta_N)$ and the chemical potential $\mu$, which have to be solved self-consistently together with the density constraint

$$\nu = \frac{1}{\mathcal{M}} \sum_{k,\alpha} \left[ \langle c^\dagger_{k\alpha\uparrow} c_{k\alpha\uparrow} \rangle + \langle c^\dagger_{-k,\alpha,\downarrow} c_{-k,\alpha,\downarrow} \rangle \right] = 1 - \frac{1}{\mathcal{M}} \sum_{k,a} n_F[E_a(k)] \sum_\alpha \left[ |\Psi_{a\alpha,-}(k)|^2 - |\Psi_{a\alpha,+}(k)|^2 \right], \tag{S16}$$

where we have introduced the total number of states $\mathcal{M} = \mathcal{N}N$ and $\nu$ is the filling factor taking into account both spin-up and spin-down electrons ($\nu = 0 \ldots 2$).

## B. Linear-response theory close to $T_c$

We may use the full set of self-consistency equations to determine the pairing amplitude $\Delta_\alpha$ at any given $T$. Close to the critical temperature $T_c$, however, we can alternatively use linear-response theory, which we derive in the following.

We start again from the BdG Hamiltonian in Eq. (S4). This time, we change to the basis of Bloch eigenstates of the normal-state Hamiltonian obtained from solving

$$\sum_\beta H_{\alpha\beta}(k)\,\psi_{n\beta}(k) = \xi_n(k)\,\psi_{n\alpha}(k), \tag{S17}$$

where $\psi_n(k)$ is the coordinate vector of the $n$-th eigenstate with energy $\xi_n(k)$ relative to the chemical potential $\mu$. The quantities $\psi_n(k)$ and $\xi_n(k)$ are typically the result of a numerical diagonalization of the Bloch Hamiltonan matrix $H(k)$. The eigenstates are normalized as $\sum_\alpha |\psi_{n\alpha}(k)|^2 = 1$. We define new creation and annihilation operators by

$$c^\dagger_{nk\uparrow} = \sum_\alpha \psi_{n\alpha}(k)\, c^\dagger_{k\alpha\uparrow} = \frac{1}{\sqrt{\mathcal{N}}} \sum_{i\alpha} \psi_{n\alpha}(k)\, e^{-ik(R_i+r_\alpha)}\, c^\dagger_{i\alpha\uparrow}, \tag{S18}$$

and by

$$c_{n,-k,\downarrow} = \sum_\alpha \psi_{n\alpha}(k)\, c_{-k,\alpha,\downarrow} = \frac{1}{\sqrt{\mathcal{N}}} \sum_{i\alpha} \psi_{n\alpha}(k)\, e^{-ik(R_i+r_\alpha)}\, c_{i\alpha\downarrow}. \tag{S19}$$

Using these definitions, the BdG Hamiltonian becomes

$$H_{BdG} = \sum_{k,n} \xi_n(k)(c^\dagger_{nk\uparrow} c_{nk\uparrow} - c_{n,-k,\downarrow} c^\dagger_{n,-k,\downarrow}) + \sum_{k,n,m} [\Delta_{nm}(k)\, c^\dagger_{nk\uparrow} c^\dagger_{m,-k,\downarrow} + h.c.], \tag{S20}$$

with

$$\Delta_{nm}(k) = \sum_\alpha \Delta_\alpha\, \psi^*_{n\alpha}(k)\, \psi_{m\alpha}(k). \tag{S21}$$

Similarly, the self-consistency equation Eq. (S6) transforms to

$$\Delta_\alpha = \frac{U}{\mathcal{N}} \sum_{k,n,m} \psi_{n\alpha}(k)\, \psi^*_{m\alpha}(k)\, \langle c_{nk\uparrow} c_{m,-k,\downarrow} \rangle. \tag{S22}$$

Introducing the alternative Nambu-space vector $\tilde{C}^\dagger_k = [c^\dagger_{1,k,\uparrow}, \ldots, c^\dagger_{N,k,\uparrow}, c_{1,-k,\downarrow}, \ldots, c_{N,-k,\downarrow}]$, we write the BdG Hamiltonian as

$$H_{BdG} = \sum_k \left[ \tilde{C}^\dagger_k \tilde{H}^0(k)\, \tilde{C}_k + \tilde{C}^\dagger_k \tilde{H}^1(k)\, \tilde{C}_k \right], \tag{S23}$$

with

$$\tilde{H}^0(k) = \text{diag}\,[\xi_1(k), \ldots, \xi_N(k), -\xi_1(k), \ldots, -\xi_N(k)], \tag{S24}$$

and

$$\tilde{H}^1(k) = \begin{pmatrix} 0 & \Delta(k) \\ \Delta^\dagger(k) & 0 \end{pmatrix} \tag{S25}$$

Next, we define an operator

$$A^{nm}(k) = c_{nk\uparrow} c_{m,-k,\downarrow} = \tilde{C}_k^\dagger \, \mathcal{A}^{nm} \, \tilde{C}_k, \tag{S26}$$

$$\mathcal{A} = \begin{pmatrix} 0 & 0 \\ Q^{nm} & 0 \end{pmatrix}, \, Q_{ij}^{nm} = -\delta_{im}\delta_{jn}. \tag{S27}$$

We can now calculate $\langle A^{nm}(k)\rangle$ using linear response theory, which yields

$$\langle A^{nm}(k)\rangle = \sum_{i \neq j} \frac{1}{\epsilon_i(k) - \epsilon_j(k)} H_{ij}^1(k) \, \mathcal{A}_{ji}^{nm} \, \{n_F[\epsilon_i(k)] - n_F[\epsilon_j(k)]\}, \tag{S28}$$

where $\epsilon_i(k) = H_{ii}^0(k)$. Using the matrices above, this expression can be simplified to

$$\langle A^{nm}(k)\rangle = \frac{1 - n_F[\xi_n(k)] - n_F[\xi_m(k)]}{\xi_n(k) + \xi_m(k)} \Delta_{nm}(k) \tag{S29}$$

Finally, by using Eqs. (S21), (S22), (S26), and (S29) we obtain the linearized self-consistency equations

$$\Delta_\alpha = \sum_\beta \chi_{\alpha\beta} \, \Delta_\beta, \tag{S30}$$

with the pair susceptibility

$$\chi_{\alpha\beta} = \frac{U}{\mathcal{N}} \sum_k \sum_{n,m} \frac{1 - n_F[\xi_n(k)] - n_F[\xi_m(k)]}{\xi_n(k) + \xi_m(k)} \psi_{n\alpha}(k)\psi_{m\alpha}^*(k)\psi_{n\beta}^*(k)\psi_{m\beta}(k). \tag{S31}$$

Here, $n, m = 1, \ldots, N$ and also $\alpha, \beta = 1, \ldots, N$, where $N$ is the number of sites per unit cell. We may determine the critical temperature $T_c$ from diagonalizing the pair susceptibility $\chi_{\alpha\beta}$: the critical temperature is reached when the largest eigenvalue of $\chi_{\alpha\beta}$ is equal to 1. The corresponding eigenvector corresponds to the normalized spatial profile $\delta_\alpha$ of the superconducting pairing amplitude at the critical temperature, i.e., $\sum_\alpha |\delta_\alpha|^2 = 1$.

For completeness, the chemical potential is determined from

$$\frac{\nu}{2} = \frac{1}{\mathcal{M}} \sum_{k,n} \langle c_{nk\uparrow}^\dagger c_{nk\uparrow}\rangle, \tag{S32}$$

Using a linear-response formula for the expectation value similar to Eq. (S28), we find that the linear-order term is zero. Therefore, we can express the expectation value directly in terms of the normal-state Fermi function (zeroth-order term), namely

$$\frac{\nu}{2} = \frac{1}{\mathcal{M}} \sum_{k,n} n_F[\xi_n(k)]. \tag{S33}$$

.

## C.  Nonlinear self-consistent equations for the magnitude of the order parameter

Determining the exact pairing amplitude $\Delta_\alpha$ at a given temperature $T < T_c$ requires self-consistently solving a set of $N + 1$ equations with $N + 1$ unknown parameters [see Eqs. (S15) and (S16)], where $N$ is the number of atoms per unit cell. Numerically, this can be done by employing a minimization algorithm. However, as the parameter space grows with $N$ such an algorithm takes long to find solutions for systems with a large number of atoms per unit cell. Since we want to study systems with large $N$, we will make use of an approximation which allows us to speed up the computations considerably.

For that purpose, we write the pairing amplitude as $\Delta_\alpha = \|\Delta\| \, \hat{\Delta}_\alpha$, where $\|\Delta\|$ is the magnitude of the vector $(\Delta_\alpha)$ and $\hat{\Delta}_\alpha$ is the normalized spatial profile of the pairing amplitude, that is $\sum_\alpha |\hat{\Delta}_\alpha|^2 = 1$. We assume that for the

systems considered the spatial profile of the order parameter is only weakly temperature-dependent such that we can write

$$\Delta_\alpha(T) \approx \|\Delta(T)\| \, \hat{\Delta}_\alpha, \tag{S34}$$

i.e., the temperature dependence of the pairing amplitude is fully given by the magnitude $\|\Delta(T)\|$. We have checked this expectation for the extended Kane-Mele model with periodic onsite disorder (see Fig. S7). We obtain a good agreement at small disorder, while the deviations are generally larger for sizeable disorder.

This motivates us to extract the spatial profile $\hat{\Delta}_\alpha$ by diagonalizing the pair susceptibility at $T_c$ as obtained from linear-response theory [see Eq. (S31)]. This allows us to reduce the set of $N+1$ self-consistency equations for $N+1$ unknown variables to a set of only two equations for $\|\Delta\|$ and $\mu$. Using Eq. (S15), we obtain

$$\|\Delta\|^2 = \sum_\alpha |\Delta_\alpha|^2 = \frac{U^2}{\mathcal{N}^2} \sum_\alpha \Big( \sum_{k,a} \Psi_{a\alpha,+}(k)\Psi^*_{a\alpha,-}(k)\{1 - n_F[E_a(k)]\} \Big)^2, \tag{S35}$$

$$\nu = 1 - \frac{1}{\mathcal{M}} \sum_{k,a} n_F[E_a(k)] \sum_\alpha \Big[ |\Psi_{a\alpha,-}(k)|^2 - |\Psi_{a\alpha,+}(k)|^2 \Big]. \tag{S36}$$

## II.  SUPERFLUID WEIGHT

### A.  Reduced BdG Hamiltonian

We consider a system with $N$ sites per unit cell, one orbital per site, periodic boundary conditions, and spin-rotation symmetry around the $z$ axis, such that the electronic part of the Bloch Hamiltonian can be block-diagonalized

$$\mathcal{H}(\mathbf{k}) = \begin{pmatrix} h_\uparrow(\mathbf{k}) - \mu & 0 \\ 0 & h_\downarrow(\mathbf{k}) - \mu \end{pmatrix}, \tag{S37}$$

where $h_\sigma(\mathbf{k})$ is the Fourier-transformed $N \times N$ hopping Hamiltonian of the electrons with spin $\sigma$, and $\mu$ is the chemical potential. Using the Nambu basis $\{c_{\alpha\mathbf{k}\uparrow}, c_{\alpha\mathbf{k}\downarrow}, c^\dagger_{\alpha,-\mathbf{k}\uparrow}, c^\dagger_{\alpha,-\mathbf{k}\downarrow}\}$, the full BdG Hamiltonian of the system takes the following general form

$$\mathcal{H}_{\mathrm{BdG}}(\mathbf{k}) = \begin{pmatrix} h_\uparrow(\mathbf{k}) - \mu & 0 & \boldsymbol{\Delta}_{\uparrow\uparrow} & \boldsymbol{\Delta}_{\uparrow\downarrow} \\ 0 & h_\downarrow(\mathbf{k}) - \mu & \boldsymbol{\Delta}_{\downarrow\uparrow} & \boldsymbol{\Delta}_{\downarrow\downarrow} \\ \boldsymbol{\Delta}^\dagger_{\uparrow\uparrow} & \boldsymbol{\Delta}^\dagger_{\downarrow\uparrow} & -h^*_\uparrow(-\mathbf{k}) + \mu & 0 \\ \boldsymbol{\Delta}^\dagger_{\uparrow\downarrow} & \boldsymbol{\Delta}^\dagger_{\downarrow\downarrow} & 0 & -h^*_\downarrow(-\mathbf{k}) + \mu \end{pmatrix}, \tag{S38}$$

with the $N \times N$ order-parameter matrices $\boldsymbol{\Delta}_{\sigma\sigma'}$, which can generally depend on $\mathbf{k}$. Here, we consider on-site interactions and assume that the pairing obeys the translation symmetry of the Hamiltonian such that the components of the order-parameter matrices are constants. The BdG Hamiltonian is particle-hole antisymmetric with the anti-unitary operator $\tau_x \otimes s_0 \otimes \mathbb{1}_{N \times N} K$, $\mathbf{k} \to -\mathbf{k}$, where $K$ is complex conjugation. This imposes the restriction $\boldsymbol{\Delta} = -\boldsymbol{\Delta}^T$ implying $\boldsymbol{\Delta}_{\downarrow\downarrow} = \boldsymbol{\Delta}_{\uparrow\uparrow} = 0$ and $\boldsymbol{\Delta}_{\uparrow\downarrow} = -\boldsymbol{\Delta}^T_{\downarrow\uparrow} \equiv \boldsymbol{\Delta}$. Therefore, the BdG Hamiltonian becomes block-diagonal after a basis transformation,

$$\tilde{\mathcal{H}}_{\mathrm{BdG}}(\mathbf{k}) = \begin{pmatrix} h_\uparrow(\mathbf{k}) - \mu & \boldsymbol{\Delta} & 0 & 0 \\ \boldsymbol{\Delta}^\dagger & -h^*_\downarrow(-\mathbf{k}) + \mu & 0 & 0 \\ 0 & 0 & h_\downarrow(\mathbf{k}) - \mu & -\boldsymbol{\Delta}^T \\ 0 & 0 & -\boldsymbol{\Delta}^* & -h^*_\uparrow(-\mathbf{k}) + \mu \end{pmatrix} \equiv \begin{pmatrix} \mathcal{H}_{\mathrm{BdG},\uparrow}(\mathbf{k}) & 0 \\ 0 & \mathcal{H}_{\mathrm{BdG},\downarrow}(\mathbf{k}) \end{pmatrix}, \tag{S39}$$

where $\mathcal{H}_{\mathrm{BdG},\sigma}(\mathbf{k})$ are the reduced BdG Hamiltonians associated with the spin-$\sigma$ blocks of electrons in the full Bloch Hamiltonian. In this basis, the particle-hole operator is $s_x \otimes \tau_x \otimes \mathbb{1}_{N \times N} K$. This implies that the two blocks are connected via

$$\mathcal{H}_{\mathrm{BdG},\uparrow}(\mathbf{k}) = -\tau_x \mathcal{H}^*_{\mathrm{BdG},\downarrow}(-\mathbf{k})\tau_x. \tag{S40}$$

This reflects the redundancy of the BdG formalism, which is why it is sufficient to focus entirely on one block, say $\mathcal{H}_{\mathrm{BdG},\uparrow}(\mathbf{k})$. Hence, from now on we will simply write $\mathcal{H}_{\mathrm{BdG}}(\mathbf{k})$ when referring to a reduced BdG Hamiltonian.

Moreover, if also time-reversal symmetry is preserved, we have $h_\uparrow(\mathbf{k}) = h_\downarrow^*(-\mathbf{k})$ and $\boldsymbol{\Delta}$ can be chosen to be real. $\boldsymbol{\Delta}$ is also diagonal, because we assumed one orbital per site and on-site interactions. Therefore, we further have $\boldsymbol{\Delta}^\dagger = \boldsymbol{\Delta}^T = \boldsymbol{\Delta}$. Hence, the reduced BdG Hamiltonian becomes

$$\mathcal{H}_{\mathrm{BdG},\uparrow}(\mathbf{k}) = \begin{pmatrix} h_\uparrow(\mathbf{k}) - \mu & \boldsymbol{\Delta} \\ \boldsymbol{\Delta} & -[h_\uparrow(\mathbf{k}) - \mu] \end{pmatrix}, \tag{S41}$$

i.e., only information from one of the spin channels enters in each BdG Hamiltonian, which allows us to effectively drop the spin index.

## B. Decomposition of the superfluid weight

The full superfluid weight of a superconductor with spin-rotation symmetry, time-reversal symmetry, and momentum-independent $\boldsymbol{\Delta}$ can be written as,

$$D_{\mu\nu}^s = \frac{e^2}{\hbar^2} \sum_{\mathbf{k},ij} \frac{n(E_j) - n(E_i)}{E_i - E_j} \left( \langle \psi_i | \partial_{k_\mu} \mathcal{H}_{\mathrm{BdG}} | \psi_j \rangle \langle \psi_j | \partial_{k_\nu} \mathcal{H}_{\mathrm{BdG}} | \psi_i \rangle - \langle \psi_i | \partial_{k_\mu} \mathcal{H}_{\mathrm{BdG}} \gamma^z | \psi_j \rangle \langle \psi_j | \gamma^z \partial_{k_\nu} \mathcal{H}_{\mathrm{BdG}} | \psi_i \rangle \right), \tag{S42}$$

where $|\psi_i(\mathbf{k})\rangle$ is the $i$-th eigenstate of the reduced BdG Hamiltonian with energy $E_i(\mathbf{k})$, $n(E_i)$ is the Fermi function, and $\gamma^z = \tau_z \otimes \mathbb{1}_{N \times N}$. For $E_j = E_i = E$ the prefactor is set to $-\partial_E n(E)$.

It is worth expressing the superfluid weight in terms of the eigenstates of the normal-state Bloch Hamiltonian $h_\sigma(\mathbf{k})$. For this purpose, we decompose the BdG eigenstates as

$$|\psi_i\rangle = \sum_{m=1}^N \left( w_{+,im} |m\rangle_\uparrow \otimes |+\rangle + w_{-,im} |m_-^*\rangle_\downarrow \otimes |-\rangle \right), \tag{S43}$$

where $|m\rangle_\uparrow$ is the eigenvector of $h_\uparrow(\mathbf{k})$ with eigenvalue $\varepsilon_{\uparrow,m}(\mathbf{k})$, $|m_-^*\rangle_\downarrow$ is the eigenvector of $h_\downarrow^*(-\mathbf{k})$ with eigenvalue $\varepsilon_{\downarrow,m}(-\mathbf{k})$, and $|\pm\rangle$ is the eigenvector of $\sigma^z$ with eigenvalue $\pm 1$. With this, the expression for the superfluid weight from Eq. (S42) becomes

$$D_{\mu\nu}^s = -2 \sum_{\mathbf{k},ij} \frac{n(E_j) - n(E_i)}{E_i - E_j} \sum_{m,n} \sum_{p,q} \left( w_{+,im}^* w_{+,jn} w_{-,jp}^* w_{-,iq} {}_\uparrow \langle m | \partial_{k_\mu} h_\uparrow(\mathbf{k}) | n \rangle_\uparrow {}_\downarrow \langle p_-^* | \partial_{k_\nu} h_\downarrow^*(-\mathbf{k}) | q_-^* \rangle_\downarrow \right.$$

$$\left. + w_{-,im}^* w_{-,jn} w_{+,jp}^* w_{+,iq} {}_\downarrow \langle m_-^* | \partial_{k_\mu} h_\downarrow^*(-\mathbf{k}) | n_-^* \rangle_\downarrow {}_\uparrow \langle p | \partial_{k_\nu} h_\uparrow(\mathbf{k}) | q \rangle_\uparrow \right). \tag{S44}$$

By defining the current operator

$$[j_{\mu,\sigma}(\mathbf{k})]_{mn} = {}_\sigma \langle m | \partial_{k_\mu} h_\sigma(\mathbf{k}) | n \rangle_\sigma = \partial_{k_\mu} \varepsilon_{\sigma,m} \delta_{mn} + (\varepsilon_{\sigma,m} - \varepsilon_{\sigma,n}) {}_\sigma \langle \partial_{k_\mu} m | n \rangle_\sigma, \tag{S45}$$

we can write the full superfluid weight as

$$D_{\mu\nu}^s = \sum_{\mathbf{k}} \sum_{m,n}^N \sum_{p,q}^N C_{pq}^{mn} \left( [j_{\mu,\uparrow}(\mathbf{k})]_{mn} [j_{\nu,\downarrow}(-\mathbf{k})]_{qp} + [j_{\nu,\uparrow}(\mathbf{k})]_{mn} [j_{\mu,\downarrow}(-\mathbf{k})]_{qp} \right), \tag{S46}$$

with

$$C_{pq}^{mn} = -2 \sum_{i,j}^{2N} \frac{n(E_j) - n(E_i)}{E_i - E_j} w_{+,im}^* w_{+,jn} w_{-,jp}^* w_{-,iq}. \tag{S47}$$

The current operator in Eq. (S45) contains two qualitatively different kinds of terms: the diagonal terms depend on the derivative of the band dispersions whereas the off-diagonal terms contain derivatives of the Bloch states. The latter, therefore, encode information about the quantum geometry of states. Accordingly, the full superfluid weight can be decomposed into a conventional and a geometric part,

$$D_{\mu\nu}^s = D_{\mathrm{conv},\mu\nu}^s + D_{\mathrm{geom},\mu\nu}^s, \tag{S48}$$

where the geometric part $D_{\mathrm{geom}}^s$ collects all contributions to the superfluid weight $D^s$ containing off-diagonal elements of the current operator, while the conventional part $D_{\mathrm{conv}}^s$ contains only diagonal elements of the current operator. The conventional part can also be written as

$$D_{\mathrm{conv},\mu\nu}^s = \sum_{\mathbf{k}} \sum_{mp}^N C_{pp}^{mm} \left[ \partial_{k_\mu} \epsilon_{\uparrow,m}(\mathbf{k}) \, \partial_{k_\nu} \epsilon_{\downarrow,p}(-\mathbf{k}) + \partial_{k_\nu} \epsilon_{\uparrow,m}(\mathbf{k}) \, \partial_{k_\mu} \epsilon_{\downarrow,p}(-\mathbf{k}) \right]. \tag{S49}$$

In this work, we compute the full superfluid weight using the general formula in Eq. (S42) and the conventional contribution to the superfluid weight based on Eq. (S49). We then compute the geometric part as the difference of the two.

## III. EXTENDED KANE-MELE MODELS

The Kane-Mele model [42] is prototypical for the realization of a quantum spin Hall insulator on a lattice. It is a

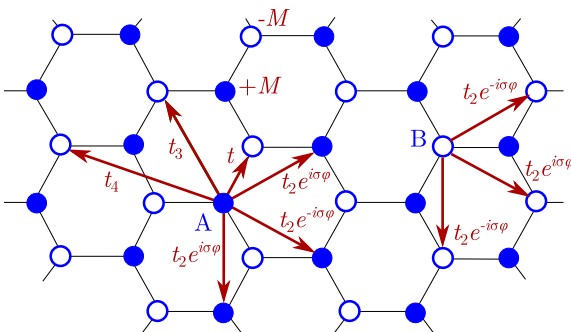

FIG. S1. Illustration of the extended Kane-Mele model on the honeycomb lattice with nearest-neighbor hopping $t$, $i$-th neighbor hopping amplitudes $t_i$, and staggered on-site potential $M$. Hopping processes between second neighbors acquire a phase $e^{\pm i\sigma\varphi}$ depending on the spin $\sigma = \pm 1 \equiv \uparrow, \downarrow$ of the involved particles, on the hopping direction, and on the sublattice A, B. Red arrows indicate a selection of hopping processes. Other hopping processes can be inferred by symmetry.

tight-binding model on the 2D honeycomb lattice, which is defined as

$$H_0 = \sum_{i,\sigma} \left[ (-1)^i M - \mu \right] c_{i\sigma}^\dagger c_{i\sigma} + t \sum_\sigma \sum_{\langle i,j \rangle_1} c_{j\sigma}^\dagger c_{i\sigma} + t_2 \sum_{\langle i,j \rangle_2} \left( e^{i\varphi_{ij}} c_{j\uparrow}^\dagger c_{i\uparrow} + e^{-i\varphi_{ij}} c_{j\downarrow}^\dagger c_{i\downarrow} \right), \tag{S50}$$

where the operators $c_{i\sigma}^\dagger$ ($c_{i\sigma}$) create (annihilate) an electron with spin $\sigma = \uparrow, \downarrow$ at site $i$, $\langle i,j \rangle_n$ denotes pairs of $n$-th neighbors, $M$ is a staggered on-site potential also known as mass term, $\mu$ is the chemical potential, and $\varphi_{ij} = \pm\varphi$ is a next-nearest-neighbor (NNN) hopping phase whose sign depends on the hopping direction, on the spin, and on the sublattice as shown in Fig. S1. Note that the phase is $\varphi = \pi/2$ in the original Kane-Mele model.

We have used the lattice vectors $\mathbf{a}_1 = (3a/2, -a\sqrt{3}/2)$ and $\mathbf{a}_2 = (0, a\sqrt{3})$. The coordinates of the two basis atoms A and B are $\mathbf{r}_A = (0,0)$ and $\mathbf{r}_A = (a/2, a\sqrt{3}/2)$, respectively, where $a$ is the distance between the two atoms. The corresponding reciprocal lattice vectors are $\mathbf{b}_1 = (4\pi/3a, 0)$ and $\mathbf{b}_2 = (2\pi/3a, 2\pi/\sqrt{3}a)$.

The model is time-reversal symmetric and block-diagonal in spin space. In particular, the two spin blocks are mapped onto each other under time reversal. Furthermore, each spin block realizes a Haldane model [43], which is a model prototypical for the realization of a Chern insulator on a lattice. The spin blocks have opposite Chern numbers $C_\uparrow = -C_\downarrow = C$, which are related to the $\mathbb{Z}_2$ topological invariant $\nu$ of the corresponding Kane-Mele model as $\nu = C \bmod 2$. For a system with spin-rotation symmetry, the latter (without mod 2) is also known as the spin Chern number. In the main text and in the following, we therefore use the Chern number $C$ of the spin-up block to specify the topology of the system.

Due to its sublattice structure and spin symmetry, the model has two spin-degenerate energy bands. At half-filling, it realizes a topological insulator with $C = \pm 1$ in certain regimes of the parameter space spanned by $(t, t_2, M, \phi)$. In particular, if $M = 0$ and $t, t_2 > 0$ we have $C = +1$ for $-\pi < \varphi < 0$ and $C = -1$ for $0 < \varphi < \pi$. On the other hand, tuning $M$ generally leads to a phase transition to a trivial insulator with $C = 0$.

By optimizing the parameters of the model, it is possible to make one of the bands quasi-flat, even in the topological phase. As a measure of the band flatness, we use the ratio of bandwidth to energy gap,

$$r = \frac{\Delta E_{\text{bandwidth}}}{\Delta E_{\text{gap}}}. \tag{S51}$$

For the Kane-Mele model defined above, at $M = 0$ the flatness $r$ of the lower band is minimal for $\cos\varphi = t/4t_2 = 3\sqrt{3/43}$. Its minimum value is about $r_{min} = 0.29$.

## A. Kane-Mele model with optimized flatness

We can make the flatness of the lower band arbitrarily small by adding further-neighbor hopping to the Kane-Mele model. Here, we go up to fourth-neighbor hopping and optimize the model parameters to minimize the flatness $r$. Our *extended* Kane-Mele model reads,

$$H = H_0 + t_3 \sum_\sigma \sum_{\langle i,j \rangle_3} c_{j,\sigma}^\dagger c_{i,\sigma} + t_4 \sum_\sigma \sum_{\langle i,j \rangle_4} c_{j,\sigma}^\dagger c_{i,\sigma}. \tag{S52}$$

For $M = 0$, the flatness of its lower band is minimal for the parameters $t_2 = 0.349t$, $t_3 = -0.264t$, $t_4 = 0.026t$, and $\varphi = 1.377$. The minimum flatness is approximately $r_{min} = 0.006$, which is about two orders of magnitude smaller than the minimum flatness of the 2nd-neigbor Kane-Mele model discussed above. In the following, we will refer to this version of the model as the "flat" Kane-Mele model.

## B. Models presented in the main text

In this subsection, we provide details on the extended Kane-Mele models presented in Fig. 1 of the main text. The cluster size of the Kane-Mele models is $8 \times 8$ clean unit cells, which is equal to 128 sites for each disordered cluster. Moreover, we used the interaction strength $U = 3t$, the filling fraction $\nu = 1/2$, and the temperature $T = T_{c,0}/100$, where $T_{c,0}$ is the critical temperature in the clean limit. The hopping parameters are fixed at the values in the flat limit.

The other parameters are as follows:

(i) topological Kane-Mele model in the flat limit: $M = 0$, $\varphi = 1.377 \equiv \varphi_{\text{opt}}$,

(ii) topological Kane-Mele model with dispersing bands: $M = t$ and $\varphi = \varphi_{\text{opt}}$,

(iii) topological Kane-Mele model with dispersing bands: $M = 0$ and $\varphi = 1.0$,

(iv) semi-metallic Kane-Mele model close to the topological phase transition: $M = 1.75t$ and $\varphi = \varphi_{\text{opt}}$,

(v) trivial Kane-Mele model with dispersing bands: $M = 3.2t$ and $\varphi = \varphi_{\text{opt}}$.

For the decomposition of the superfluid weight into conventional and geometric contributions as shown in Fig. 2 of the main text, we have used smaller clusters of size $5 \times 5$ corresponding to 50 sites within each disordered cluster.

To generate all the tight-binding Hamiltonians with disorder we have used the software package Kwant [47].

## C. Flat Kane-Mele model with disorder

We study flat Kane-Mele models with random onsite disorder and periodic boundary conditions. The random onsite potentials are drawn from a uniform distribution on the interval $[-W, W]$ with the disorder strength $W$. For the Kane-Mele models considered here, unless stated otherwise, the disordered cluster has a size of $8 \times 8$ unit cells of the clean system, which amounts to 128 sites.

Generally, we find that the pairing amplitude $\Delta$ is suppressed by disorder. We use this observation to define a disorder scale $W_0$ through $\langle \bar{\Delta} \rangle (W_0) = \Delta_0/2$, where $\bar{\cdot}$ denotes a sample average and $\langle \cdot \rangle$ a disorder average. $\Delta_0$ is the sample average of the pairing amplitude in the clean limit. Numerically, we determine $W_0$ by linear interpolation based on the set of disorder strengths $W$ considered. For instance, for $8 \times 8$ clusters of the flat Kane-Mele model we obtain $W_0 = 1.52t$, while we get $W_0 = 1.85t$ for smaller clusters of size $5 \times 5$.

In Fig. S2, we present various properties of the flat Kane-Mele model as a function of the disorder strength $W$. Figure S2(a) shows disorder-averaged spectral properties. We observe that the band gap between the lower and the upper band decreases linearly with disorder and closes around $W = 2W_0$. Beyond this value, the gap slowly opens again. On the contrary, the bandwidth of the previously flat lower band increases linearly.

We further find, see Fig. S2(b) that the disorder-averaged superfluid weight $\langle D_s \rangle$ is proportional to the disorder- and sample-averaged pairing amplitude $\langle \bar{\Delta} \rangle$ in the small-disorder regime $W \ll W_0$, i.e.,

$$\langle D_s \rangle = \frac{D_{s,0}}{\Delta_0} \langle \bar{\Delta} \rangle, \tag{S53}$$

where the proportionality constant is given by the ratio of the respective values in the clean limit, $D_{s,0}$ and $\Delta_0$.

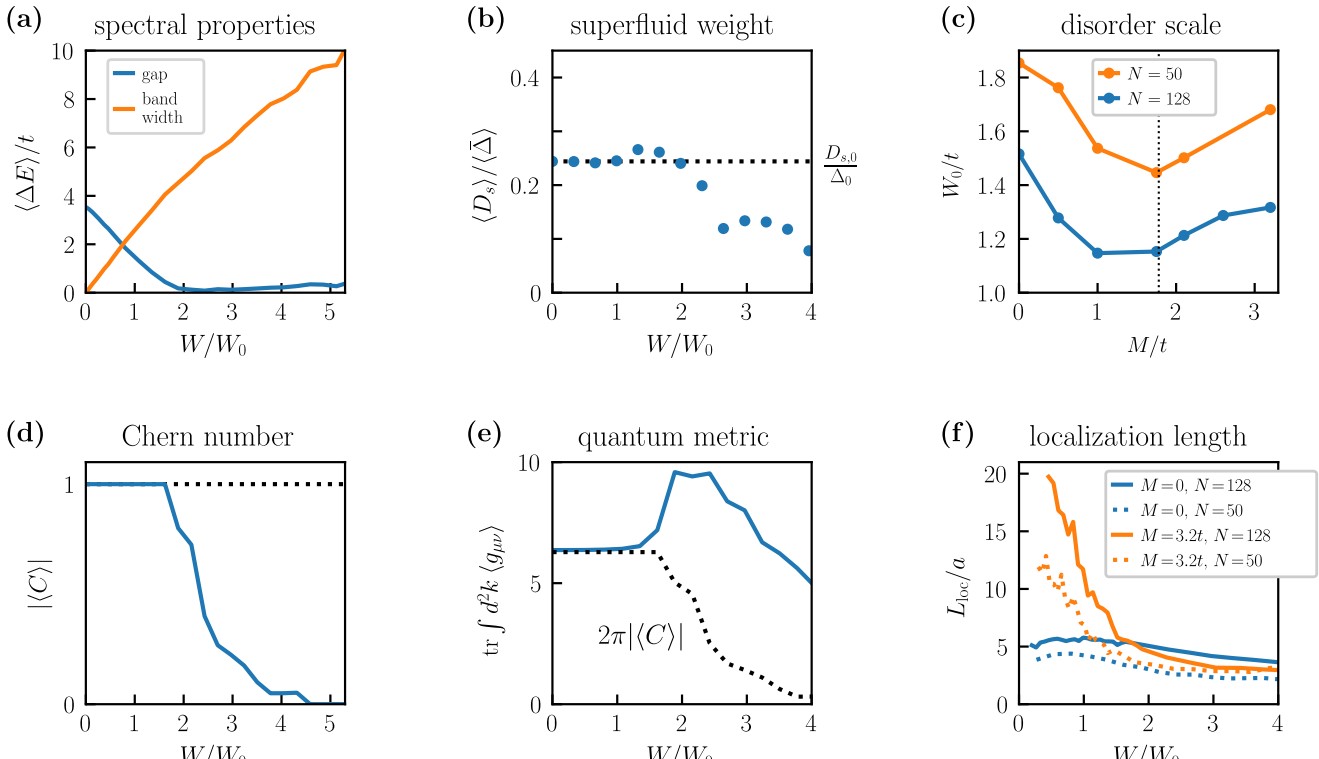

FIG. S2. Disordered Kane-Mele model with optimized flatness: (a) Evolution of the band gap and the bandwidth of the lower band as a function of disorder $W$. (b) Relation between full superfluid weight $D_s$ and sample-averaged superconducting order parameter $\bar{\Delta}$. (c) Disorder scale $W_0$ as a function of the staggered on-site potential $M$ for two different system sizes $N$. The dotted black line indicates the topological transition in the clean system. (d) Evolution of the Chern number of the lower spin-up band. (e) Evolution of the trace of the quantum metric of the lower spin-up band integrated over the Brillouin zone. The dotted black line indicates the evolution of the lower bound on this integral given by the Chern number $C$. (f) Localization length as a function of the disorder parameter $W$ for the flat ($M = 0$) and for a trivial ($M = 3.2t$) Kane-Mele model for two different system sizes $N$. The localization length is given in units of the sublattice separation $a$ of the honeycomb lattice.

In Fig. S2(c), we show the disorder scale $W_0$ as a function of the staggered on-site potential for the two system sizes considered in this work, namely $N = 50$ ($5 \times 5$ cluster) and $N = 128$ ($8 \times 8$ cluster).

In the clean limit, the spin -up flat band of the considered model has Chern number $C = -1$. As expected, the value of the Chern number is robust as long as the energy gap remains open, as we show in Fig. S2(d). Once the disorder-averaged gap becomes close to zero, more and more realizations within the disorder ensemble undergo a transition to a trivial phase. Hence, the absolute value of the disorder-averaged Chern number decreases until it reaches $\langle C \rangle = 0$. At this point, the gap is large enough such that all realizations have undergone the transition from topological to trivial.

We have also computed the quantum metric of the model adopting the essence of a method for calculating the Berry curvature in a discretized Brillouin zone [48] to efficiently compute the quantum geometric tensor $\mathcal{B}_{ij}$. In Fig. S2(e), we show the evolution of the trace of the disorder-averaged quantum metric $g_{\mu\nu}$ integrated over the Brillouin zone. In the main text, we stated that this quantity is bounded from below by $2\pi|C|$ in the clean limit, where $C$ is the Chern number of the involved band. Here, we find that this bound also applies to the averages in the disordered model.

In Fig. S2(f), we relate the disorder strength $W$ to the localization length $L_{\mathrm{loc}}$ of the system. For this purpose, we compute the average two-terminal conductance $G$ for a ribbon of fixed width as a function of its length $L$ for different disorder $W$ using periodically repeating clusters of size $8 \times 8$. For disorder strengths $W > 0.1t$, we find that the conductance shows a clear exponential suppression with a saturation $G_\infty$:

$$G(L) = C e^{-L/L_{\mathrm{loc}}} + G_\infty. \tag{S54}$$

By fitting our numerical results to this expression, we extract the localization length $L_{\mathrm{loc}}(W)$. In Fig. S2(f), we present our results for the flat topological Kane-Mele model ($M = 0$) and for a trivial Kane-Mele model ($M = 3.2t$)

for two different system sizes $N$.

## D.   Kane-Mele models with different staggered on-site potential $M$

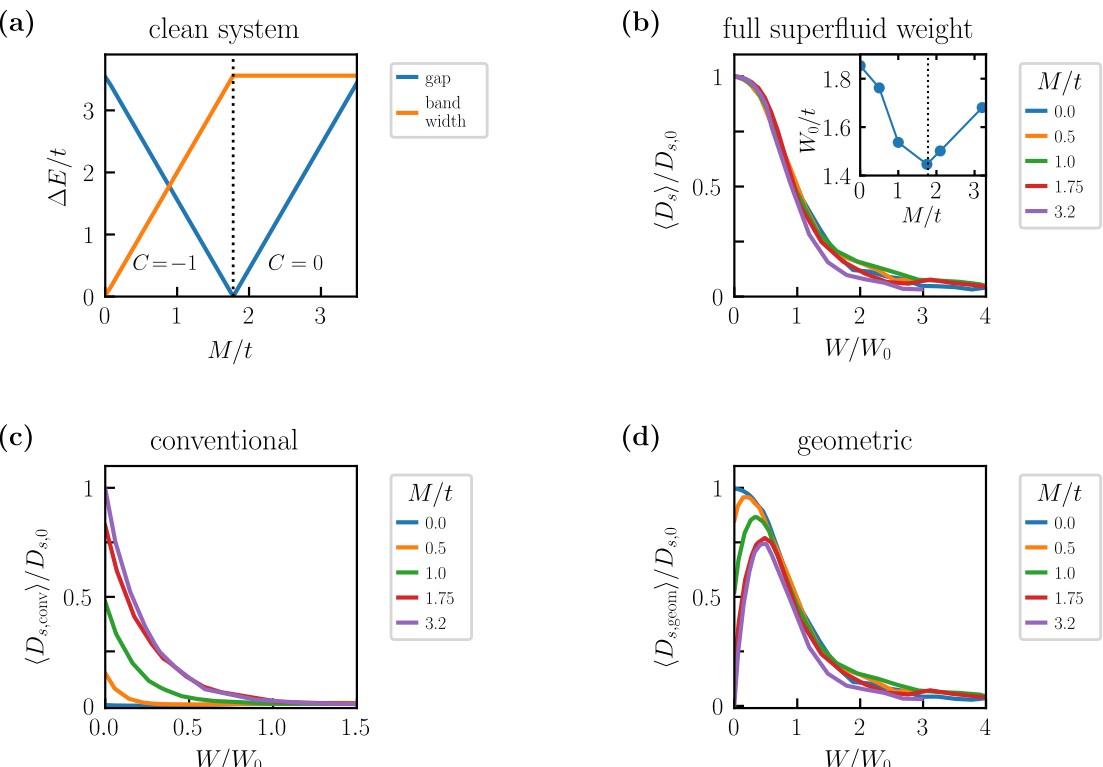

FIG. S3.    Properties of the extended Kane-Mele model as a function of the staggered onsite potential $M$ for $U = 3t$, $T = T_{c,0}/100$, and $\nu = 1/2$: (a) Evolution of the energy gap and the bandwidth of the lower band. We also indicate the Chern number of the lower spin-up band on the two sides of the topological transition. Full superfluid weight (b), conventional contribution (c), and geometric contribution (d) as a function of disorder $W$ for various values of $M$. The inset in (b) shows the disorder scale $W_0$ as a function $M$.

By changing the staggered on-site potential $M$, we can tune the system from the topological to the trivial phase. In the following, we discuss how the change of topology affects the full, the conventional, and the geometric superfluid weight as disorder is increased. Since the decomposition of the superfluid weight into its two contributions is numerically more involved, we here present results for smaller clusters of size $5 \times 5$ clean unit cells amounting to 50 sites per cluster. All other model parameters are the same as in the flat Kane-Mele model.

In Fig. S3(a), we first show how the bandwidth of the lower band and its gap to the upper band changes as a function of $M$ in the clean system. The band gap decreases linearly, closes at about $M = 1.78t$, and then increases again linearly. In the process, the Chern number of the lower spin-up band changes from $C = -1$ to $C = 0$ indicating a transition from a topological to a trivial insulator. In contrast, the bandwidth grows linearly until the gap closing point and remains constant after that.

Figure S3(b) shows the full superfluid weight as a function of disorder for different $M$. We study the system at filling $\nu = 1/2$ and interaction parameter $U = 3t$. Comparing to Fig. S3(a), we find that the suppression of the superfluid weight is independent of the topology of the underlying band. It follows a universal behavior as discussed in the main text.

In contrast to that, the conventional and geometric contributions presented in Figs. S3(c) and (d), respectively, show a clear dependence on the parameter $M$. In the flat limit ($M = 0$), the conventional contribution approximately vanishes independent of disorder, such that the superfluid weight remains entirely geometric. With increasing $M$, the fraction of the conventional contribution grows in the small disorder regime $W \ll W_0$, while the fraction of

the geometric contribution is suppressed. This is attributed to the increasing bandwidth. Moreover, once the band acquires a finite dispersion the geometric contribution shows a non-monotonic behavior in the small disorder regime with a maximum shifting to larger disorder values with increasing $M$. However, in the large disorder regime $W \gg W_0$ the conventional contribution vanishes and the superfluid weight becomes again entirely geometric.

### E. Kane-Mele models with different NNN hopping phases $\varphi$

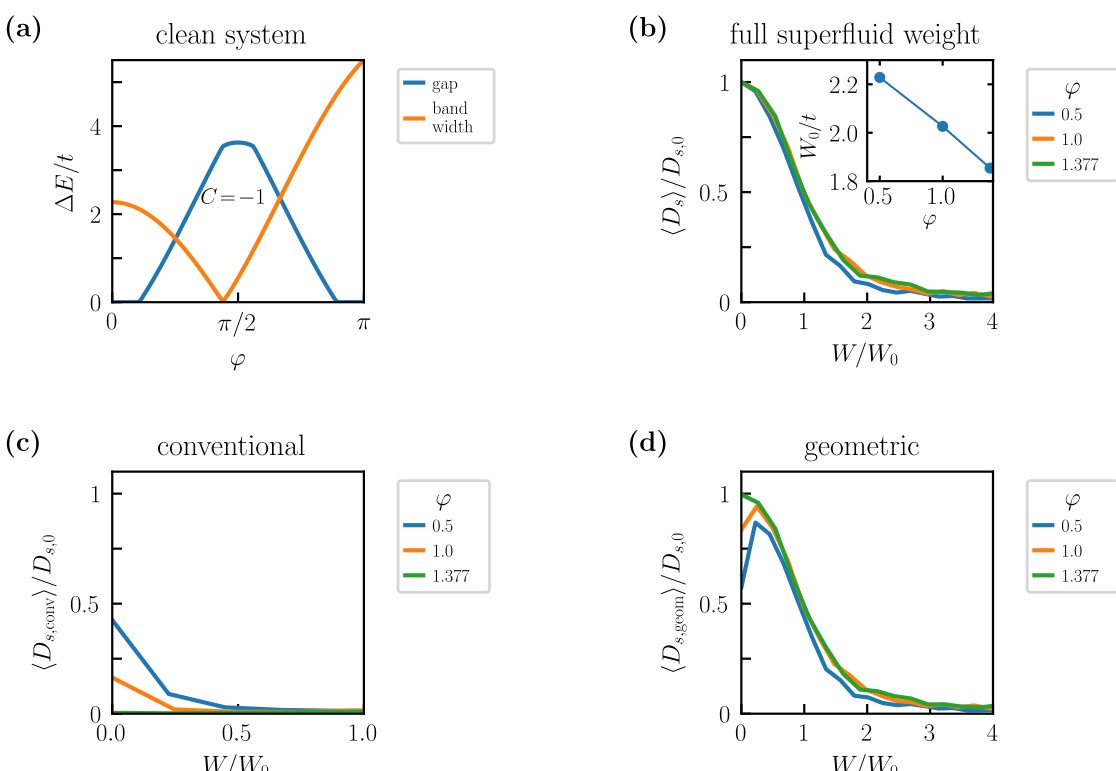

FIG. S4. Properties of the extended Kane-Mele model as a function of the NNN hopping phase $\varphi$ for $U = 3t$, $T = T_{c,0}/100$, and $\nu = 1/2$: (a) Evolution of the energy gap and the bandwidth of the lower band. The band has optimal flatness at $\varphi_{\mathrm{opt}} = 1.377$. We also indicate the Chern number of the lower spin-up band where the gap is nonzero. Full superfluid weight (b), conventional contribution (c), and geometric contribution (d) as a function of disorder $W$ for various fixed NNN hopping phases $\varphi$. The inset in (b) shows the disorder scale $W_0$ as a function of $\varphi$.

Next, we discuss how a change of the NNN hopping phase affects the behavior of the superfluid weight and its two contributions under disorder. Again, for numerical reasons we present results for a smaller cluster of size $5 \times 5$. All other parameters are the same as in the flat Kane-Mele model.

First of all, starting from the flat limit with $M = 0$, we note that a change of the NNN hopping phase cannot push the clean system into the trivial phase. As we show in Fig. S4(a), the energy gap closes close to $\varphi = 0$ and close to $\varphi = \pi$. Under time reversal, we have that $\varphi \to -\varphi$ and $C \to -C$. Hence, tuning $\varphi$ into the interval $[-\pi, 0]$ the energy gap opens again and the Chern number of the lower spin-up band changes from $C = -1$ to $C = 1$. The system remains a topological insulator. As expected, the bandwidth increases away from the flat limit with $\varphi_{\mathrm{opt}} = 1.377$.

Turning to the behavior of the superfluid weight, we make the similar observations as in the case of varying the $M$ parameter. Again, we study the system at filling $\nu = 1/2$ with interaction parameter $U = 3t$. The full superfluid weight shows a universal behavior independent of the value of the NNN hopping phase [see Fig. S4(b)]. On the contrary, the conventional and geometric contributions show a clear parameter dependence in the small disorder regime $W \ll W_0$ [see Fig. S4(c) and (d)]. In particular, as the bandwidth of the underlying band becomes sizeable the conventional contribution is enhanced whereas the geometric contribution is suppressed in this regime. For large disorder, the superfluid becomes entirely geometric independent of the NNN hopping phase.

In contrast to what we observe for the variation of $M$, we find that the disorder scale $W_0$ *increases* approximately linearly as we tune the system from the flat limit to the band closing point close to $\varphi = 0$, see inset of Fig. S4(b).

## IV.   STANDARD DEVIATIONS AND FORMATION OF SUPERCONDUCTING ISLANDS

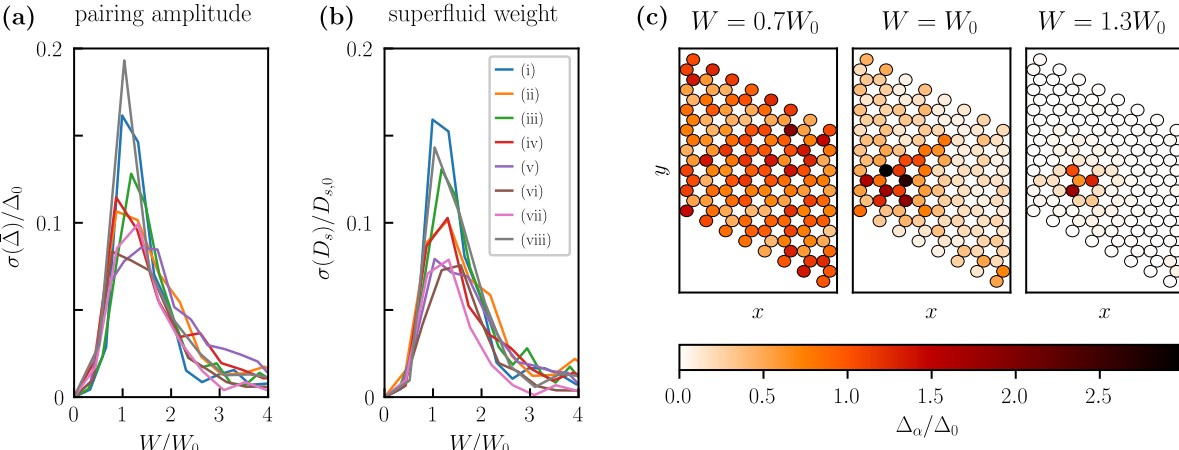

FIG. S5.   Ensemble standard deviations $\sigma$ of (a) the pairing amplitude $\bar{\Delta}$ (spatial average) and (b) the superfluid weight $D_s$ as a function of $W/W_0$ for the models considered in Fig. 1 of the main text: (i)-(v) topological and trivial extended Kane-Mele models, (vi)-(viii) trivial single-band models. (c) Spatial profile $\Delta_\alpha$ of the pairing amplitude for single disorder realizations of the flat Kane-Mele model [model (i)] at different disorder strengths $W/W_0$. White corresponds to a vanishing pairing amplitude, whereas darker colors signify a larger values. Around $W = W_0$, the superconductor breaks up into isolated superconducting islands accompanied by large fluctuations of $\bar{\Delta}$ and $D_s$.

In the main text, we pinned down a universal suppression of the pairing amplitude $\Delta$ and of the superfluid weight $D_s$ reflected in the ensemble averages across various models. In this section, we show that this universality applies widely also to the fluctuations around the average. For this purpose, we analyze the standard deviations, $\sigma(A) = \sqrt{\langle A^2 \rangle - \langle A \rangle^2}$, of the respective quantities for the models discussed in Fig. 1 of the main text.

Figures S5(a) and (b) show our results for the spatial average of the pairing amplitude and for the superfluid weight, respectively. We find that fluctuations reach a maximum at $W \simeq W_0$, where $W_0$ is the disorder scale introduced in the main text. Quantitatively, we observe that the behavior at small and large disorder is model-independent, which is in line with the universality of the ensemble averages. On the contrary, the height of the fluctuation peaks at $W \simeq W_0$ is found to be model dependent, but we are not able to identify any systematic signature originating from the topology of the models as both the flat Kane-Mele model [model (i)] and the trivial single-band model [model (viii)] can have comparable peak fluctuations.

A closer inspection of the spatial profile of the pairing amplitude for single disorder realizations reveals that the the fluctuations are maximal ($W \simeq W_0$) when the superconductor starts to break up into superconducting islands [see Fig. S5(c)]. For $W/W_0 > 1$ the superconducting order parameter $\Delta_\alpha$ vanishes in large parts of the sample except inside small, isolated clusters. We observe the breaking of the superconductor into superconducting island around $W \simeq W_0$ in all the models considered, providing a concrete physical interpretation for the disorder scale $W_0$.

## V.   SIZE SCALING OF UNIVERSAL SUPPRESSION

In the main text, we found that the superfluid weight shows a universal and model-independent suppression by disorder. In this section, we show how this universal behavior evolves as a function of the cluster size. Due to the universality, we restrict our analysis to a specific model, namely the extended Kane-Mele model with a flat lower band.

Figure S6 shows our results for samples with different numbers of sites $N$ in the disordered supercell. Both the sample-averaged pairing amplitude [Fig. S6(a)] and the superfluid weight [Fig. S6(b)] show a saturation at large

**(a)**  pairing amplitude

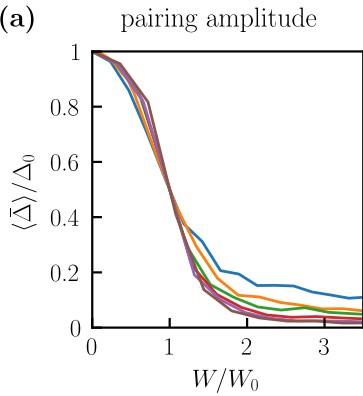

**(b)**  superfluid weight

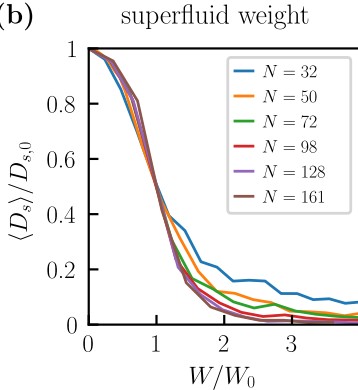

FIG. S6. Disorder-induced suppression of the pairing amplitude and the superfluid weight for different cluster sizes. We show results for the extended Kane-Mele model with flat lower band and $U = 3t$, $T = T_{c,0}/100$, and $\nu = 1/2$.

disorder strengths $W/W_0$. This value is finite for small cluster sizes but approaches zero as the cluster size is increased. In the small-disorder regime, the considered quantities are slightly enhanced as the cluster size is increased but approach a common functional behavior once the cluster size is sufficiently large. Overall, we find that the functional form of the disorder-induced suppression does not change considerably beyond cluster sizes of $N = 128$ for the considered model. In general, we expect that the required cluster size that approximates well the $N \to \infty$ behavior depends on the details of the system.

## VI.   PAIRING AMPLITUDE OBTAINED FROM FULL MEAN-FIELD EQUATIONS

**(a)**  topol. ($M = 0$)

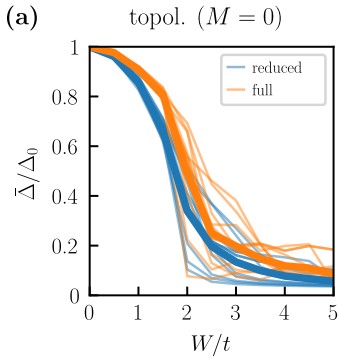

**(b)**  trivial ($M = 3.2t$)

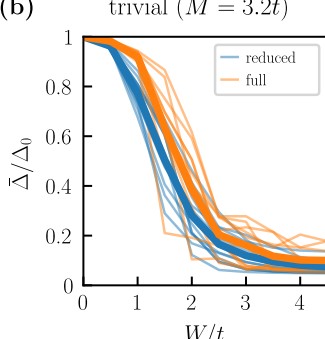

FIG. S7. Comparison between full (orange) and reduced (blue) mean-field approach for the superconducting order parameter $\Delta$ in the extended Kane-Mele model with $U = 3t$, $T = T_{c,0}/100$, and $\nu = 1/2$: (a) topological phase and (b) trivial phase. We show results of single disorder realizations (thin lines) and their ensemble averages (bold lines).

So far, we have used a *reduced* set of mean-field equations to self-consistently determine the real-space structure of the pairing amplitude [see Eqs. (S36) and (S36)]. In this section, we also look at the solutions of the full mean-field equations (S15) and (S16) for the extended Kane-Mele model with disorder in the zero-temperature limit.

For that purpose, we generate several disorder realizations for a disordered cluster of size $N = 50$ for different disorder strengths $W$. We then solve both the reduced and the full mean-field equations self-consistently. We use the solutions of the reduced mean-field equations as initial guesses for the solver algorithm searching for solutions of the full mean-field equations. Figure S7 shows the spatial averages of the different realizations and also their ensemble averages. For both the trivial and topological phase of the model, we find a good agreement between both approaches at small disorder. At larger disorder, the discrepancies become more enhanced with the solutions of the reduced mean-field equations tending to overestimate the suppression of the pairing amplitude. Nevertheless, the differences in the

ensemble averages remain small thereby justifying our approximation. Most importantly, the qualitative behavior of the suppression is the same in both approaches. In particular, also the solutions of the full mean-field equations do not show any significant difference between the topological and the trivial phase.

We note that the solutions of the full mean-field equations obtained here are not necessarily the solutions with the lowest free energy. We find that the obtained solutions are sensitive to the initial conditions used for the solver, which indicates a highly complex structure of the corresponding free-energy landscape with many local minimums. Therefore, finding a global minimum based on the full mean-field equations is computationally very expensive. Due to their significantly smaller parameter space, the reduced mean-field equations provide a computationally more efficient and more robust approach to finding suitable solutions. The analysis in this section, as well as additional calculations with different initial conditions, leads us to the expectation that this does not affect the results qualitatively. Therefore, we have used the reduced mean-field equations in the rest of the text.

## VII.   TRIVIAL SINGLE-BAND MODELS

In the main text we compare our results obtained for the extended Kane-Mele model to a trivial single-band model defined on a 2D square lattice. The model has one orbital per site and is described by the Hamiltonian

$$H = -t \sum_{\sigma} \sum_{<i,j>} c_{j\sigma}^{\dagger} c_{i\sigma} - \mu \sum_{\sigma,i} c_{i\sigma}^{\dagger} c_{i\sigma}. \tag{S55}$$

It has a single spin-degenerate energy band with dispersion $E = -2t(\cos k_x + \cos k_y)$. Each spin band is entirely trivial, i.e., all components of the quantum geometric tensor are zero in the whole Brillouin zone. Hence, their quantum metric, Berry curvature, and Chern number are zero as well.

To make this single-band model comparable to the flat Kane-Mele model, we use the full superfluid weight of the latter in the clean limit, $D_{s,0} = 0.2245\, t_{\mathrm{KM}}$ where $t_{\mathrm{KM}}$ is the nearest-neighbor hopping amplitude of the flat Kane-Mele model, as a common energy scale. We then generate a set of models with different hopping parameters $t$, interaction strengths $U$, and fillings $\nu$, such that, in the clean limit, they all have the same superfluid weight $D_{s,0}$.

For the trivial single-band models in Fig. 1 of the main text, we have used clusters of size $11 \times 11$, which is equal to 121 sites per disordered cluster. Moreover, the presented models have the following parameters:

 (vi) $U = 13.4\, D_{s,0}$, $\nu = 1$, and $t = 2.0\, D_{s,0}$,

 (vii) $U = 8.9\, D_{s,0}$, $\nu = 1$, and $t = 1.7\, D_{s,0}$,

(viii) $U = 13.4\, D_{s,0}$, $\nu = 1/5$, and $t = 3.3\, D_{s,0}$.

For the decomposition of the superfluid weight into conventional and geometric contributions as shown in Fig. 3 of the main text, we have used smaller clusters of size $7 \times 7$ corresponding to 49 sites within each disordered cluster.

To generate the tight-binding Hamiltonians with disorder we have used the software package Kwant [47].

## VIII.   SUPERFLUID WEIGHT FOR CLEAN SYSTEMS

For a conventional superconductor originating from a metallic state given by a partially-filled, isolated, and approximately parabolic band, the superfluid weight is purely conventional and can be expressed as

$$D_s = e^2 \frac{n}{m^*}, \tag{S56}$$

with the electron density $n$ and the effective mass $m^*$. On the other hand, for a superconductor originating from a metallic state given by a partially-filled and isolated flat band, the superfluid weight is entirely geometric and is related to the quantum geometry of the electronic states as

$$[D_s]_{ij} = \frac{8e^2}{\hbar^2} \Delta \sqrt{\nu(1-\nu)} \int \frac{d^d k}{(2\pi)^d}\, g_{ij}(\mathbf{k}), \tag{S57}$$

where $\Delta$ is the superconducting order parameter, $\nu$ the band filling, $d$ the dimensions, and $g_{ij}(\mathbf{k}) =$ is the quantum metric. The latter is obtained as the real part of the quantum geometric tensor $\mathcal{B}_{ij}(\mathbf{k}) = \langle \partial_i u_{n\mathbf{k}} | \left(1 - |u_{n\mathbf{k}}\rangle\langle u_{n\mathbf{k}}|\right) | \partial_j u_{n\mathbf{k}}\rangle$, with $|u_{n\mathbf{k}}\rangle$ the Bloch functions of the flat band. The superconducting order parameter further satisfies

$$\Delta = \frac{U}{2} \sqrt{\nu(1-\nu)}. \tag{S58}$$

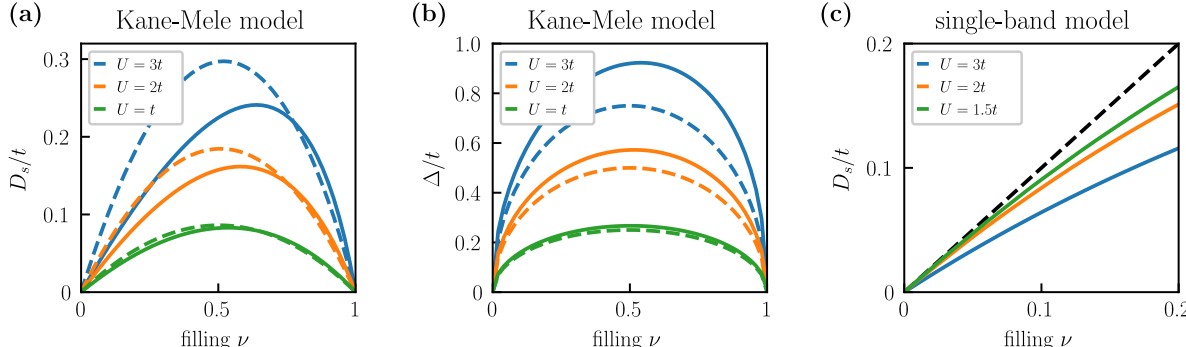

FIG. S8. Comparison with analytical formulas for the clean systems in the zero-temperature limit: (a) superfluid weight $D_s$ and (b) superconducting order parameter $\Delta$ of the flat Kane-Mele model for different coupling constants $U$ as a function of the filling $\nu$. Solid lines represent the respective quantities as obtained directly from numerics, dashed lines show the results computed using Eqs. (S57) and (S58), respectively. (c) Superfluid weight of the single-band model for different $U$ at small fillings close to the band bottom. Solid lines are the numerical results while the dashed black line represents the analytical result using Eq. (S56) assuming a parabolic band.

Importantly, the flat-band formulas are expected to hold for coupling constants $U$ much smaller than the excitation energy to the other bands and much larger than the bandwidth of the flat band. In the following, we apply the analytical formulas above to the models considered in this paper in the clean limit.

We first look at the extended Kane-Mele model in the flat limit. The flat lower band has a bandwidth of $0.02t$ and is separated from the upper band by an energy gap of $3.5t$, where $t$ is the first-neighbor hopping defining the energy scale of the model. In Fig. S8(a) we show the superfluid weight of the model (solid lines) as a function of the filling $\nu$ for different coupling constants $U$. We further compare this to the results of Eq. (S57) (dashed lines). For the latter, we have computed the quantum metric of the model numerically adopting the essence of a method for calculating the Berry curvature in a discretized Brillouin zone [48] to efficiently compute the quantum geometric tensor $\mathcal{B}_{ij}$. At larger coupling constants $U$, we observe that the numerically obtained curve for the superfluid weight is skewed with respect to the analytical result. However, with decreasing $U$ the agreement improves. Around $U = t$, the two curves are nearly on top of each other. This is in agreement with the validity regime of Eq. (S57).

We have also checked the superconducting order parameter of this model as a function of $U$ [see Fig. S8(b)]. Also here we observe deviations from the analytical formula in Eq. (S58) at larger $U$, but the agreement improves as the coupling constant is decreased to be considerably smaller than the energy gap of the model.

Finally, we look at the superfluid weight of the single-band model. Close to the band bottom at $\Gamma$, the model has an approximately parabolic dispersion with an associated effective mass of $m^* = \hbar^2/2ta^2$, where $a$ is the lattice constant and $t$ the nearest-neighbor hopping. We further have $n = \nu/a^2$ for the electron density. Hence, Eq. (S56) evaluates to $D_s = 2e^2\nu t/\hbar^2$ for our specific model, which is plotted in Fig. S8(c) for small band fillings $\nu$ (dashed line). We also plot the numerically obtained superfluid weight for different coupling strengths $U$ (solid lines). We find that, as $U$ decreases, the superfluid weight of the model approaches the analytical result at small band fillings.