# Peer review of "Universal suppression of superfluid weight by disorder independent of quantum geometry and band dispersion"

_SciPost Physics_

## Round 1 · Referee Report · Anonymous (Referee 1) · 2022-6-2

Strengths

  1. Clear and concise.
  2. Timely

Weaknesses

  1. Title of the work (see report).
  2. Much of the numerical work is, in some sense, somewhat limited. The systems sizes are on the smaller side, have very large attractive interactions, and the superconducting self-consistency equations are only solved approximately by using only the linear Tc solutions with a self-consistent amplitude at lower temperatures. Ideally, one would have like to see more robust numerical work, but on the other hand, I do not really think that the results would change if included, especially give how close the present results align with previous numerical works [10.1103/PhysRevB.65.014501,10.1103/PhysRevLett.81.3940]. The authors also explicitly verify that their approach is valid, at least for moderately weak disorder, in section 5 and 6 of the supplementary. I therefore contend that this is not a problem, but a small yellow flag to keep in mind for the strongly discorded case.

Report

The idea behind this work is timely and interesting. During the last decade, there has been an increasing recognition that the all-important superfluid weight of superconductors can have non-conventional geometric contributions in multiband systems, meaning that the superfluid weight can be non-zero even for flat bands, which is important for systems such as twisted bilayer graphene but also more generally for our understanding of superconductors.

The unconventional origin of the geometric superfluid weight contributions does however also lead to new questions. In this work the authors ask how the average superfluid weight changes with an increasing (on-site, Anderson) disorder strength in several systems and for different system parameters. The work is predominately numerical and has strong parallels to Ref.36 of the manuscript [10.1103/PhysRevLett.81.3940], but unlike that work, this work updates our understanding by also separating out the conventional and geometric parts of the superfluid weight. Their motivation is sound, for as the authors say; the geometric superfluid weight has to be finite when the electronic bands have a non-trivial quantum geometry, bounded below by for instance the Chern number. A-priori one could therefore expect that the behavior in topological systems (with finite Chern number) should be different from the case of a standard one-band metals superconductor.

The results of this work are however surprising. Rather than a distinctive difference between the topological and trivial regimes of the extended Kane-Mele model that they study, they find that the suppression of the average superfluid weight (as well as the standard deviation) follows the same relationship in all cases when normalized by their initial values (a simple square lattice model serves as a control).

I therefore believe that these unexpected findings will stimulate further work, given the current strong interest in these newly recognized geometrical contributions. For these reasons, I however also think that the title of this work might be a problem, as it might overstate the claims of this work. Importantly, the present work only considers on-site s-wave superconductivity and one type of disorder. One might reasonably ask just how "universal" the present results therefore are, or if the results might change for more general systems. I therefore think the authors should think about the title to see if another title may better reflect exactly what is already shown in this work.

With an eye towards the Sci-Post's General Acceptance Criteria, the work is concise and well written. Together with the detailed supplementary, the work should also be easy to reproduce. I find that the most relevant literature is also cited and that the material is available in the Zenodo repository. I therefore recommend publication.
  • validity: good
  • significance: high
  • originality: good
  • clarity: top
  • formatting: excellent
  • grammar: excellent

Author:  Alexander Lau  on 2022-08-11  [id 2721]

(in reply to Report 1 on 2022-06-02)
Category:
answer to question

The referee writes:

The idea behind this work is timely and interesting. During the last decade, there has been an increasing recognition that the all-important superfluid weight of superconductors can have non-conventional geometric contributions in multiband systems, meaning that the superfluid weight can be non-zero even for flat bands, which is important for systems such as twisted bilayer graphene but also more generally for our understanding of superconductors. The unconventional origin of the geometric superfluid weight contributions does however also lead to new questions. In this work the authors ask how the average superfluid weight changes with an increasing (on-site, Anderson) disorder strength in several systems and for different system parameters. The work is predominately numerical and has strong parallels to Ref.36 of the manuscript [10.1103/PhysRevLett.81.3940], but unlike that work, this work updates our understanding by also separating out the conventional and geometric parts of the superfluid weight. Their motivation is sound, for as the authors say; the geometric superfluid weight has to be finite when the electronic bands have a non-trivial quantum geometry, bounded below by for instance the Chern number. A-priori one could therefore expect that the behavior in topological systems (with finite Chern number) should be different from the case of a standard one-band metals superconductor. The results of this work are however surprising. Rather than a distinctive difference between the topological and trivial regimes of the extended Kane-Mele model that they study, they find that the suppression of the average superfluid weight (as well as the standard deviation) follows the same relationship in all cases when normalized by their initial values (a simple square lattice model serves as a control). I therefore believe that these unexpected findings will stimulate further work, given the current strong interest in these newly recognized geometrical contributions.

Our response: We thank the referee for the positive report, for putting our work in context, and for acknowledging the timeliness and novelty of our work.

The referee writes:

For these reasons, I however also think that the title of this work might be a problem, as it might overstate the claims of this work. Importantly, the present work only considers on-site s-wave superconductivity and one type of disorder. One might reasonably ask just how "universal" the present results therefore are, or if the results might change for more general systems. I therefore think the authors should think about the title to see if another title may better reflect exactly what is already shown in this work.

Our response: We thank the Referee for this comment. We understand the Referee's concerns regarding the potentially misleading generality of the title. Therefore, we have made the title more specific to reflect the fact that we consider non-magnetic disorder and $s$-wave superconductors. The new title is "Universal suppression of superfluid weight by non-magnetic disorder in $s$-wave superconductors independent of quantum geometry and band dispersion". We have also added a clarifying sentence in the abstract.

The referee writes:

Much of the numerical work is, in some sense, somewhat limited. The systems sizes are on the smaller side, have very large attractive interactions, and the superconducting self-consistency equations are only solved approximately by using only the linear Tc solutions with a self-consistent amplitude at lower temperatures. Ideally, one would have like to see more robust numerical work, but on the other hand, I do not really think that the results would change if included, especially give how close the present results align with previous numerical works [10.1103/PhysRevB.65.014501,10.1103/PhysRevLett.81.3940]. The authors also explicitly verify that their approach is valid, at least for moderately weak disorder, in section 5 and 6 of the supplementary. I therefore contend that this is not a problem, but a small yellow flag to keep in mind for the strongly discorded case.

Our response: We thank the Referee for pointing this out and for acknowledging the validity of our approach despite giving us the small yellow flag. We agree with the Referee that the interaction strengths are large and the system sizes are on the smaller side, but the consistency with earlier results suggests that heavier numerical calculations are not expected to change our results. We have used reasonably large interaction strengths because it allows us to obtain reliable results also with reasonably small system sizes, making it easier to perform a comprehensive and systematic study of conventional and geometric contributions using several different models. The study of the disorder-induced suppression of the superfluid weight at very small interaction strengths requires significantly heavier numerics (or advanced analytical techniques), and therefore we leave it as an interesting challenge for future research.

The referee writes:

With an eye towards the Sci-Post's General Acceptance Criteria, the work is concise and well written. Together with the detailed supplementary, the work should also be easy to reproduce. I find that the most relevant literature is also cited and that the material is available in the Zenodo repository. I therefore recommend publication.

Our response: We thank the Referee for their positive assessment of our work and for recommending publication. We are confident that the changes in the manuscript are in line with this assessment.

---

## Round 1 · Referee Report · Anonymous (Referee 2) · 2022-7-13

Strengths

1-Timely.
2-Careful analysis and universal behaviour found.
2-Detailed supplemental material together with code, which allows for the reproduction of the results and extensions.

Weaknesses

1-As pointed out by another referee, the title is a bit of an overstatement of the results.
2-It is not very clear from the analysis why there is a universal suppression.

Report

The authors analyze the effect of quantum geometry on the robustness to disorder in the superfluid weight of superconductors. This matter is analyzed for an extended Kane-Mele model, which presents flat bands, and a trivial single band hopping model, both with s-wave superconducting pairing. They obtain the surprising result that the scaled pairing amplitude and superfluid weight show a universal dependence with disorder, regardless of the flatness of the bands, although the conventional and geometric contributions behave in different ways.
The authors did a very careful and thorough analysis of the system and provided both very detailed Supplemental Material and the code used to generate the results. Therefore, this work has the prospect of generating interesting future expansions. I found it particularly interesting that the trace of the quantum metric increases, at least for some range of disorder, with disorder strength. It is also a timely analysis considering the recent massive interest in superconductivity in systems with flatbands.
I believe that the manuscript already fulfils SciPost criteria of publication, but there are some (optional) points that the authors can address that maybe improve the manuscript (see below).

Requested changes

These are optional points that maybe improve the manuscript. I suggest that at least 1, 4, and 5 are considered.

  1. Although the scaled $\Delta$ and $D_s$ present a universal behaviour, $W_0$ depends on $M$ and, I believe, also $\Delta_0$ and $D_{s0}$ should change depending on the parameters of the Kane-Mele extended model. Thinking about practical applications of the results, it can be beneficial to show a version of Fig. 1 without the scaling but with the quantities in units of $t$, for instance.
  2. Can one read the universality reported in Figs. 1 and S6 as a scaling property? The data collapse reminds me very much of what is observed close to a quantum critical point.
  3. When the system is disordered, translation invariance is broken. I understand that considering disorder average effectively restores translation invariance, but it is not clear whether the use of expressions of the quantum metric and the superfluid weight that rely on the use of bands is justified. Is there a definition of these quantities in real space, and do they provide the same ensemble-averaged results?
  4. Concerning the writing of the manuscript, some symbols are used for different quantities throughout the text. Although it is usually clear what they refer to depending on the context, some redefinition of variables can avoid confusion. $\nu$ is used for both spatial index and filling (this is especially confusing in (1) where they appear together). $\mu$ is used for both spatial index and chemical potential. $i$ and $j$ are used for degrees of freedom that are not spin, band indexes, and supercell labels.
  5. I believe there is a typo above (S58). The inline equation $g_{ij}(\mathbf{k})=$ seems to be missing its right-hand side.

  • validity: good
  • significance: high
  • originality: high
  • clarity: top
  • formatting: good
  • grammar: perfect

Author:  Alexander Lau  on 2022-08-11  [id 2722]

(in reply to Report 2 on 2022-07-13)
Category:
answer to question

The referee writes:

The authors analyze the effect of quantum geometry on the robustness to disorder in the superfluid weight of superconductors. This matter is analyzed for an extended Kane-Mele model, which presents flat bands, and a trivial single band hopping model, both with s-wave superconducting pairing. They obtain the surprising result that the scaled pairing amplitude and superfluid weight show a universal dependence with disorder, regardless of the flatness of the bands, although the conventional and geometric contributions behave in different ways. The authors did a very careful and thorough analysis of the system and provided both very detailed Supplemental Material and the code used to generate the results. Therefore, this work has the prospect of generating interesting future expansions. I found it particularly interesting that the trace of the quantum metric increases, at least for some range of disorder, with disorder strength. It is also a timely analysis considering the recent massive interest in superconductivity in systems with flatbands.

Our response: We thank the Referee for carefully reading our manuscript, highlighting our main results, and for the positive assessment.

The referee writes:

I believe that the manuscript already fulfils SciPost criteria of publication, but there are some (optional) points that the authors can address that maybe improve the manuscript (see below). These are optional points that maybe improve the manuscript. I suggest that at least 1, 4, and 5 are considered.

Our response: We thank the Referee for affirming that the criteria for publication in SciPost are fulfilled. We appreciate the Referee's suggestions, which have helped us to improve our manuscript. Below, we provide a point-by-point response to all suggestions.

The referee writes:

1 - Although the scaled $\Delta$ and $D_s$ present a universal behaviour, $W_0$ depends on $M$ and, I believe, also $\Delta_0$ and $D_{s,0}$ should change depending on the parameters of the Kane-Mele extended model. Thinking about practical applications of the results, it can be beneficial to show a version of Fig. 1 without the scaling but with the quantities in units of $t$, for instance.

Our response: We thank the Referee for this suggestion. Indeed, also $\Delta_0$ and $D_{s,0}$ (and also $W_0$) change with the model parameters, as already indicated in Fig. S8 of Sec. VIII in the Appendix for the flat Kane-Mele model. To provide the reader with similar data also for the other models considered in this work, as suggested by the Referee, we have added another version of Fig. 1 of the main text to the new Sec. IX of the Appendix (see new figure Fig. S9), where we show the corresponding quantities in units of $t$ without rescaling.

The referee writes:

2 - Can one read the universality reported in Figs. 1 and S6 as a scaling property? The data collapse reminds me very much of what is observed close to a quantum critical point.

Our response: We thank the Referee for this interesting question. We do not think that the universality we report is related to a quantum critical point, because we observe it also far away from any phase transitions. Nevertheless, more detailed analytical understanding of the data collapse is certainly an interesting (and challenging) direction for future research.

The referee writes:

3 - When the system is disordered, translation invariance is broken. I understand that considering disorder average effectively restores translation invariance, but it is not clear whether the use of expressions of the quantum metric and the superfluid weight that rely on the use of bands is justified. Is there a definition of these quantities in real space, and do they provide the same ensemble-averaged results?

Our response: To our knowledge, there exist no real-space expressions for the quantum metric, geometric superfluid weight, and conventional superfluid weight. The lack of such expressions is the motivation for our numerical approach where we choose a large disordered supercell and repeat it periodically in space. This way, the existing momentum-space formulas for quantum metric, geometric superfluid weight, and conventional superfluid weight can be applied. We note that our size-scaling analysis suggests that the used supercell is sufficiently large in our calculations. Nevertheless, we agree with the Referee that it would be very useful to develop a real-space approach for calculation of the geometric and conventional superfluid weight.

The referee writes:

4 - Concerning the writing of the manuscript, some symbols are used for different quantities throughout the text. Although it is usually clear what they refer to depending on the context, some redefinition of variables can avoid confusion. $\nu$ is used for both spatial index and filling (this is especially confusing in (1) where they appear together). $\mu$ is used for both spatial index and chemical potential. $i$ and $j$ are used for degrees of freedom that are not spin, band indexes, and supercell labels.

Our response: We thank the Referee for pointing this out. For the sake of clarity, we have tried to stick to standard nomenclature used in the literature, such as $\mu$ for the chemical potential, $\nu$ for the filling factor, or $g_{\mu\nu}$ for the quantum metric. Unfortunately, this inevitably leads to a double use of certain symbols. We have tried to avoid confusion by providing enough context wherever a certain symbol is used. However, we agree that especially the use of the symbol $\nu$ both as the filling factor and as an index in Eq. (1) might cause confusion. We have therefore introduced the symbol $\bar{\nu}$ throughout the text to denote the filling factor. Furthermore, we now consistently use $g_{\mu\nu}$ and $D^{\mu\nu}$ for the quantum metric and for the superfluid weight tensor.

The referee writes:

5 - I believe there is a typo above (S58). The inline equation $g_{ij}(k) =$ seems to be missing its right-hand side.

Our response: Indeed. We thank the Referee for pointing this out. We have corrected the typo by removing the "$=$" sign before $g_{ij}(k)$. The quantum metric $g_{ij}(k)$ is defined in the sentence after that. As mentioned in the response above, we have changed $g_{ij}$ to $g_{\mu\nu}$ for notational consistency.

Additional weaknesses pointed out by the Referee:

The referee writes:

As pointed out by another referee, the title is a bit of an overstatement of the results.

Our response: We understand both referees' concerns regarding the choice of title. Therefore, we have made the title more specific to reflect the fact that we consider non-magnetic disorder and $s$-wave superconductors. The new title is "Universal suppression of superfluid weight by non-magnetic disorder in $s$-wave superconductors independent of quantum geometry and band dispersion". We have also added a clarifying sentence in the abstract.

The referee writes:

It is not very clear from the analysis why there is a universal suppression.

Our response: Besides the numerical results that we present, showing how the disorder affects differently the conventional and geometric part of the superfluid weight resulting in universal suppression of the total superfluid weight, we were not able to obtain analytical expressions that could provide more insights on the mechanism for such universal behavior. This is an interesting (and challenging) direction for future research.

We thank the Referee for their positive assessment of our work and for recommending publication. We are confident that the changes in the manuscript are in line with this assessment.

---

## Editorial Decision

resubmitted